# Capsules and their traits shape phage susceptibility and plasmid conjugation efficiency

Matthieu Haudiquet[1,2] ✉, Julie Le Bris [1,3], Amandine Nucci[1], Rémy A. Bonnin [4,5,6], Pilar Domingo-Calap [7], Eduardo P. C. Rocha [1,8] ✉ & Olaya Rendueles [1,8] ✉

Bacterial evolution is affected by mobile genetic elements like phages and conjugative plasmids, offering new adaptive traits while incurring fitness costs. Their infection is affected by the bacterial capsule. Yet, its importance has been difficult to quantify because of the high diversity of confounding mechanisms in bacterial genomes such as anti-viral systems and surface receptor modifications. Swapping capsule loci between *Klebsiella pneumoniae* strains allowed us to quantify their impact on plasmid and phage infection independently of genetic background. Capsule swaps systematically invert phage susceptibility, revealing serotypes as key determinants of phage infection. Capsule types also influence conjugation efficiency in both donor and recipient cells, a mechanism shaped by capsule volume and conjugative pilus structure. Comparative genomics confirmed that more permissive serotypes in the lab correspond to the strains acquiring more conjugative plasmids in nature. The least capsule-sensitive pili (F-like) are the most frequent in the species' plasmids, and are the only ones associated with both antibiotic resistance and virulence factors, driving the convergence between virulence and antibiotics resistance in the population. These results show how traits of cellular envelopes define slow and fast lanes of infection by mobile genetic elements, with implications for population dynamics and horizontal gene transfer.

Bacterial capsules are one of the outermost cellular structures. They protect from multiple challenges such as desiccation, bacteriophage (phage) predation[1,2] protozoan grazing[3], and the host's immune cells like macrophages[4–6]. Their ability to protect the cell may explain why they are an important virulence factor among all nosocomial species, including the ESKAPE pathogens *Enterococcus faecium*, *Staphylococcus*

*aureus*, *Klebsiella pneumoniae*, *Acinetobacter baumannii*, *Pseudomonas aeruginosa*, and *Enterobacter spp*[7]. Around half of the bacterial genomes encode at least one capsule[8], usually composed of thick, membrane-bound polysaccharide polymers surrounding the cell[9]. Group I capsules, also known as Wzx/Wzy-dependent capsules, are the most frequent ones[8]. They share the assembly pathways but have very

[1]Institut Pasteur, Université Paris Cité, CNRS UMR3525, Microbial Evolutionary Genomics, Paris 75015, France. [2]Ecole Doctorale FIRE–Programme Bettencourt, CRI, Paris, France. [3]Sorbonne Université, Collège Doctoral, Ecole Doctorale Complexité du Vivant, 75005 Paris, France. [4]Team Resist UMR1184 Université Paris Saclay, CEA, Inserm, Le Kremlin-Bicêtre, Paris, France. [5]Service de bactériologie, Hôpital Bicêtre, Université Paris Saclay, AP-HP, Le Kremlin-Bicêtre, Paris, France. [6]Centre National de Référence Associé de la Résistance aux Antibiotiques, Le Kremlin-Bicêtre, Paris, France. [7]Instituto de Biología Integrativa de Sistemas, Universitat de València-CSIC, 46980 Paterna, Spain. [8]These authors contributed equally: Eduardo P. C. Rocha, Olaya Rendueles. ✉e-mail: matthieu.haudiquet@gmail.com; eduardo.rocha@pasteur.fr; olaya.rendueles-garcia@pasteur.fr

diverse sets of enzymes which result in a large variety of capsule compositions, called capsule serotypes[10]. Their highly variable nature suggests that capsule composition is under some sort of balancing or diversifying selection[11]. Accordingly, capsule loci are frequently lost and acquired by horizontal gene transfer (HGT)[12,13].

Phage infections have an important impact on bacterial population dynamics[14,15]. They require an initial step of adsorption of the viral particle to the host's cell surface. Since the capsule covers the latter, phage-capsule interactions are key determinants of the success of phage infections. In many cases phages are blocked by the obstacle caused by the capsule and cannot reach their cell receptor[1,16]. Yet, some phages have evolved mechanisms to use the capsule as a receptor or just to bypass it, such as encoding hydrolases called depolymerases which digest the capsule[17–22]. Such phages can recognize and attach to capsules of the serotype for which they encode a depolymerase. Hence, the phage-bacteria antagonistic co-evolutionary process has made these phages dependent on the bacterial capsule to adsorb efficiently to the cell. This explains why the capsule serotype shapes the host range of these phages[12,23,24]. In *K. pneumoniae*, a species where most strains are heavily capsulated, the tropism of phages to one or a few serotypes results in an excess of successful infections between strains of the same serotype[12,23,24]. This could explain why selection for *K. pneumoniae* resistant to phages often results in mutants where the capsule is inactivated[15,22,23,25] or swapped to another serotype by HGT[11,12,26,27]. There is thus a complex interplay between bacteria and phages in species where most strains are capsulated: phages are blocked by the capsule, when it hides the cell surface and they lack appropriate depolymerases [6,18], or they depend on the presence of a capsule when they can adsorb and depolymerise it[27–29]. This interplay has implications for population dynamics as the lytic cycle of most phages results in cell death. It also affects genetic exchanges because temperate phages can integrate their genome within the bacterial one and provide novel traits by lysogenic conversion. Many phages also drive the horizontal transfer of bacterial DNA by transduction[30].

Conjugation is the other main mechanism of HGT driven by mobile genetic elements (MGEs)[31]. It plays a key role in bacterial adaptation by mediating the transfer of many traits, from secondary metabolism to antibiotic resistance[32–34]. It relies on mating pair formation (MPF) systems that include a type IV secretion system. The latter were classed into eight distinct types based on gene content and evolutionary history[35]. The three most prevalent MPF types of plasmids in Proteobacteria are $MPF_F$ (named after plasmid F), $MPF_T$ (named after the Ti plasmid) and $MPF_I$ (named after the IncI R64 plasmid)[36]. While $MPF_F$ conjugative pili are typically long (up to 20 μm)[37,38], flexible and retractable[39], $MPF_T$ and $MPF_I$ are shorter and rigid (<1 μm)[39–42]. Even though there is very little published information on the effect of capsules on conjugation, we showed recently that conjugation of one plasmid from a non-capsulated *Escherichia coli* strain to different *Klebsiella* strains was higher in Δ*wcaJ* mutants which do not produce capsules than in the capsulated wild type strains[12]. Similar effects of *wcaJ* inactivation were later found for one $MPF_T$ plasmid[43] and $bla_{KPC-2}$-carrying plasmids of undetermined types[44]. However, the mechanism underlying these results remains unknown. Furthermore, conjugation rates might be more affected by certain serotypes or combinations of serotypes[45]. Computational studies failed to find an excess of conjugation between strains of the same serotype relative to pairs of strains with different ones[12], but whether some serotypes have systematic higher conjugation rates is unknown. Similarly, there is little information on how the presence of a capsule in the donor cell affects conjugation.

*K. pneumoniae* is a good model to study the effect of capsules on the transfer of MGEs. Indeed, most strains have one single genetic locus encoding a groupI or Wzx/Wzy-dependent capsule, which varies a lot in terms of chemical composition between serotypes (of which

more than 130 have been characterized computationally in this species)[46–48]. So far, eight distinct monosaccharides have been identified in the 77 chemically characterized serotypes, as well as many modifications such as acetylation and a myriad of glycosidic bonds[46]. *K. pneumoniae* is an important nosocomial pathogen in which many virulence factors and antibiotic resistance genes are encoded in plasmids including carbapenemases and 16 S rRNA methylases[49–52]. It is also a bacterial species with a large environmental breadth outside mammals[53,54]. *K. pneumoniae* genomes contain many prophages, which upon induction have been shown to produce temperate phages that are specific of one or a few serotypes[23]. It is also the focus of very productive lines of research to develop phage therapies to target multi-resistant strains[55]. The clinical characteristics of *K. pneumoniae* are associated with specific serotypes. Some clones are called hyper-virulent because they provoke infections in healthy individuals including liver abscesses. They are almost exclusively of the K1 or K2 serotypes[56] and often produce very thick capsules that are thought to facilitate infection of humans[57]. Other examples include K3 strains which are associated with rhinoscleroma, or K24 which are associated with nosocomial infections. Both K3 and K24 are often multi-drug resistant because they acquired multiple MGEs encoding antibiotic resistance genes[58,59]. Hence, virulence and resistance are associated with specific serotypes and both traits were gained by the acquisition of conjugative elements encoding them[60]. Yet, the interplay between capsules and conjugation and how this affects bacterial traits remains poorly understood.

Here, we sought to characterize the influence of capsule serotypes on the infection of *K. pneumoniae* cells by phages and conjugative plasmids. As mentioned above, previous works showed that phage host range depends on the capsule serotype[12,17,23,24] whereas plasmid acquisition might be facilitated by the loss of the capsule[12,43,44]. Yet, it has remained difficult to isolate the impact of these effects because strains with similar serotypes also tend to be more genetically related. A precise understanding of the interplay between capsules and MGEs needs a control for variation of strains' genetic background because the success and rates of infections also depend on the cell physiology (e.g., growth rate), the presence of defence systems (e.g., restriction-modification), the envelope composition (e.g., LPS), and genetic interactions between the MGE and the genome (e.g., repression of incoming phages by resident prophages)[61,62]. To solve this deadlock, we swapped capsular loci among strains. This is challenging because capsule swapping requires to build complex isogenic mutants, i.e., precisely exchanging ~30 kb capsule loci between strains. We devised a scalable method to generate such *K. pneumoniae* serotype swaps and used it to study phage and plasmid infection in strains with different serotypes in isogenic backgrounds. These mutants allowed to show that serotype swaps lead to a swap of phage host range. We then leveraged a diverse set of clinically relevant plasmids to test the importance of the capsule in shaping conjugation efficiency in the recipient cell. The control for the genetic background allowed us to show that capsules also affect conjugation efficiencies of the donor cell and that some serotypes are associated with higher rates of transfer. To shed light on the mechanisms explaining these results, we quantified the effective volume of individual cells in bacterial colonies and tested whether the volume of the capsule affects conjugation efficiency. Finally, we used comparative genomics to test if these results contribute to explain the distribution of plasmids in hundreds of complete genomes of *K. pneumoniae*. Indeed, the frequency of the different types of plasmids and the number of plasmids recently acquired per serotype match the expectations given by the experimental data.

## Results
### Serotype swap shifts phage sensitivity and resistance
To test the precise influence of the capsule and its different serotypes on infection by MGEs, we first deleted the complete capsule locus

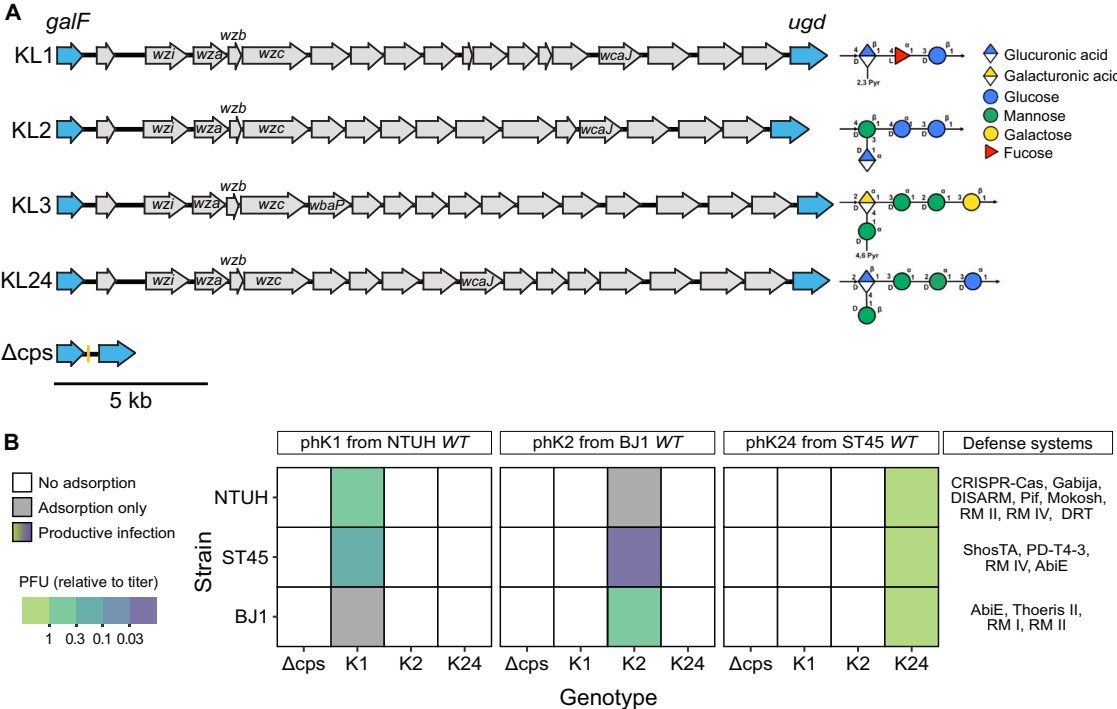

**Fig. 1 | Capsule serotype determines phage infectivity. A** Overview of the genetic loci encoding the four different capsule serotypes included in this study and their chemical composition. Arrows represent the different genes, *galF* and *ugd* in blue corresponding to the regions involved in homologous recombination to generate the swaps. Conserved genes involved in assembly and export of the capsule (*wza*, *wzb*, *wzc*, *wzi*) and initiating glycosyl-transferase (*wcaJ*, *wbaP*) are labelled. The chemical composition of the capsule (monomers and their organisation), is displayed on the right of each locus (predicted by K-PAM[129]). **B** Matrix of phage infection. Infection assay for each of the three phages (panels), three swapped

(Δcps) and confirmed that the mutants were non-capsulated (Supplementary Fig. S1). We focused on a set of three strains recently isolated, belonging to three different sequence types (ST), namely NTUH (ST23), BJ1 (ST380), and an ST45 isolate (see Dataset S1 *Strains*). We used these strains as chassis to insert the four selected serotypes (serotypes K1, K2, K3 and K24 (Fig. 1A) via a novel scarless method (See *Construction of mutants*). These serotypes were chosen for their clinical relevance (see above) and because the genetics and the chemical composition of their capsules are well-known[46]. All mutants were verified by whole-genome sequencing (See *Mutants and transconjugants validation*). These null and swapped mutants allow us to assess the interplay between MGEs, capsules (and their serotypes) while controlling for the host genetic background.

In parallel, we generated lysates from three virulent phages, each able to replicate in only one of the three wildtype strains. For clarity, we refer to those phages according to the capsule serotype of their original host: phK1, phK2 and phK24. As expected, the Δcps mutants were resistant to all three phages, indicating that all phages required the presence of a capsule for infection (Fig. 1B). We then enquired if the host sensitivity to phages is also lost when the serotype is swapped. We challenged the mutants having swapped capsular loci with the phages. We found that all those chassis strains that were susceptible to the phages became resistant upon change of the serotype (Fig. 1B) showing that capsule inactivation or swapping are sufficient to make bacteria resistant to phages for which they were originally sensitive.

If capsule serotypes are the key determinants of phage host range, then upon a capsule serotype swap, one should observe an inversion of sensitivity. We tested if the acquisition of a novel serotype led to a gain of sensitivity to the cognate phage. Indeed, in four out of six cases the

strains (y-axis), and different genotypes (x-axis). White tiles correspond to non-productive infection, i.e., no plaque could be identified. Coloured tiles correspond to the average PFU/mL normalized by the lysate titre for productive infections (Supplementary Fig. S2). Grey tiles correspond to non-productive infection with significant adsorption, while white tiles correspond to non-adsorptive pairs (Supplementary Fig. S3). Values are the mean of three independent replicates after log₁₀-transformation. Strain-specific defence systems identified by DefenseFinder as of 02/2023[63] are displayed on the right. Source data are provided as a Source Data file 1 (Adsorption) and Source Data file 2 (Infection).

serotype swap resulted in the emergence of sensitivity to phages to which the wildtype strain was resistant (Fig. 1B, S2). To understand the two cases (BJ1::K1+phK1 and NTUH::K2+phK2) not resulting in productive infections, we performed adsorption assays and observed significant adsorption at the cell surface for both pairs, in the same proportion as for susceptible hosts (Figure S3) proving that these phages can adsorb to the cell when the latter expresses the cognate capsule. We considered two hypotheses for why infection is subsequently hampered in these cases. First, the phages could depend on a secondary receptor. In this case, our results indicate that the capsule is the primary receptor, as phages are unable to adsorb to the Δcps mutants of susceptible hosts. Second, anti-phage systems in the hosts could interfere with phage infection after adsorption. To enquire on this possibility, we analysed the genomes of the three strains with DefenseFinder[63,64] and found that each strain encoded four to eight known defence systems with no homologs in the other strains (Fig. 1B). The presence of the defence systems or the possible absence of a secondary receptor may explain why phages sometimes adsorb without starting productive infections. Overall, these results show that serotype swaps result in the inversion of the patterns of sensitivity to phages, even if sometimes gaining a novel capsular locus does not allow productive phage infection.

## The serotype of the recipient influences conjugation
To identify the effects of the capsule (and its serotype) in the cell's ability to receive conjugative plasmids while controlling for the genetic background, we assembled and sequenced a collection of 10 diverse conjugative plasmids from clinical isolates (See *Selection and characterization of conjugative plasmids*) belonging to mating-pair

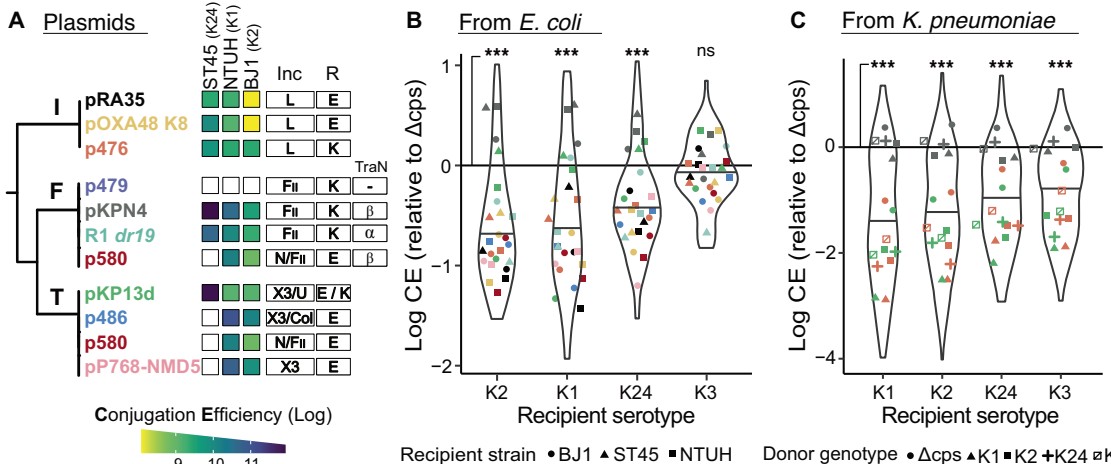

**Fig. 2 | Recipient's capsule and serotype influences conjugation efficiency.** **A** Conjugative plasmids included in the analysis. The plasmids are presented on a cladogram representing the evolutionary relations between the MPF types[36]. Note that p580 encodes two separate MPF systems of type F and T, but the F-type locus is interrupted by a transposon. Plasmid names colours match the colours of the points in the other panels. The three first columns correspond to the mean conjugation efficiency ($n = 3$) measured from *E. coli* to each of the three wild type strains. Additionally, we indicate the predicted incompatibility (Inc) groups, the antibiotic used for selection (R), either Ertapenem (E) or Kanamycin (K), and the TraN allele for F-type plasmids. **B** $\log_{10}$-transformed conjugation efficiency (CE) relative to the associated Δcps mutant (by subtraction of the $\log_{10}$-transformed

values) by capsule serotype of the recipient, from *E. coli* DH10B donors. Points represent the mean of independent triplicates after $\log_{10}$-transformation and subtraction, with colours and shapes corresponding respectively to plasmids (**A**) and strains. Solid line at $y = 0$ represents the conjugation efficiency of Δcps mutant. We used paired Wilcoxon tests (two-sided) to assess statistically significant differences. **C** Same as (**B**), but with *K. pneumoniae* strains as donors. The shapes of the data points represent the genotype of the donor strain. Source data are provided as a Source Data file 3 (Conjugation from *E. coli*) and Source Data file 4 (Conjugation from *K. pneumoniae*). Panels showing data of the individual biological replicates are presented in Supplementary Figs. S4 and S5. \*\*\*$p < 0.001$; ns $p > 0.05$.

formation type F, T and I and with diverse incompatibility groups (Fig. 2A). Of note, MPF$_F$ plasmids harboured different TraN alleles (Fig. 2A) which were recently shown to interact with surface receptors of recipient cells[65]. We set up an experimental design for plasmid conjugation including a short surface mating assay on lysogeny broth (LB) nutrient pads. The short period of time given to conjugation avoids the interference of potential differences in growth rates between donors and recipients, as well as having to account for transconjugants as donors[66,67]. We then computed the values of conjugation efficiency[66,67] (Supplementary Figs. S4, S5, S6) (See *Conjugation assays*).

We performed a first set of assays (set E1) from a non-capsulated *E. coli* strain to the *K. pneumoniae* strains, including 10 plasmids, three recipient strains, and five capsule states (Δcps, K1, K2, K3, K24, Supplementary Fig. S4). The assays were done in triplicate, accounting for a total of 450 independent conjugation experiments. We used *E. coli* DH10B as a donor because it can stably propagate large plasmids and it lacks defence systems, other plasmids, or even prophages, and is readily selectable. We observed measurable conjugation events for all pairwise strain-plasmid combinations, except for p479$_F$ in all three strains, and pP768-NMD5$_T$, p486$_T$ and p580$_{T/F}$ in strain ST45 (Fig. 2A). When measurable, conjugation from an *E. coli* donor was significantly lower in capsulated strains than in the Δcps mutants in three out of four serotypes (Fig. 2B, Wilcoxon tests, all $p < 0.01$). This confirms and generalises our previous results using one single plasmid and Δ*wcaJ* mutants[12].

In a second set of experiments (set E2), we wished to understand the patterns of conjugation within the *K. pneumoniae* species, while controlling for the genetic background (Supplementary Fig. S5). We selected three plasmids – p476$_I$, pKPN4$_F$ and pKP13d$_T$ – which are from different MPF types and efficiently transferred from and to our three *K. pneumoniae* chassis strains (Fig. 2A). We recovered the *K. pneumoniae* transconjugants of these plasmids and used them as donors to perform conjugation assays between all combinations of donors and recipients among the five capsule states (Δcps, K1, K2, K3, K24) in a

total of 675 experiments. The capsulated strains of the four serotypes had significantly lower efficiencies of acquisition of plasmids by conjugation than the Δcps strains (Fig. 2C, Wilcoxon test all $p < 0.001$). Of note, the conjugation efficiencies of the pKPN4 plasmid alone were not significantly different between capsulated and non-capsulated cells (Paired Wilcoxon test, $p > 0.05$). These results confirm that expression of a capsule is generally associated with lower rates of plasmid acquisition.

We then assessed if plasmid acquisition by conjugation is affected by the capsule serotype of the recipient cell. Considering all experiments (E1 and E2), pairwise comparisons between conjugation efficiencies across the combinations of serotypes showed that they were different for every pair of serotypes (Pairwise Wilcoxon tests, all $p < 0.01$). We used this information to rank the serotypes from lower to higher median conjugation efficiencies, resulting in the following hierarchy: K1 < K2 < K24 < K3. We also observed that the impact of the recipient capsule serotype is dependent on the plasmid. This is especially evident for plasmid pKPN4$_F$ whose conjugation into capsulated strains was as efficient as into Δcps strains (grey points, Fig. 2B, C). This was not due to higher intrinsic conjugation efficiency relative to the other plasmids, since pKPN4$_F$ displayed a median efficiency between pKP13d$_T$ and p476$_I$ in E2 (Supplementary Fig. S6D). However, plasmid pKPN4$_F$ carried the TraNβ allele (Fig. 2A), which recognizes *K. pneumoniae* OmpK36 L3 loop region[65]. We found that our three isolates encoded the *ompK36* gene with an intact L3 loop (See *TraN and TraN receptors typing*), leading to the hypothesis that TraN-receptor interactions may alleviate the impact of the capsule. Hence, the capsule negatively affects the acquisition of conjugative plasmids, with its quantitative effect depending on the specific serotype and on the plasmid.

### Donor's serotype influences conjugation

There is no available information on the effect of capsules on the frequency of conjugation by the donor cell. To test the impact of the donor serotype on conjugation efficiency, we analysed our assays relative to

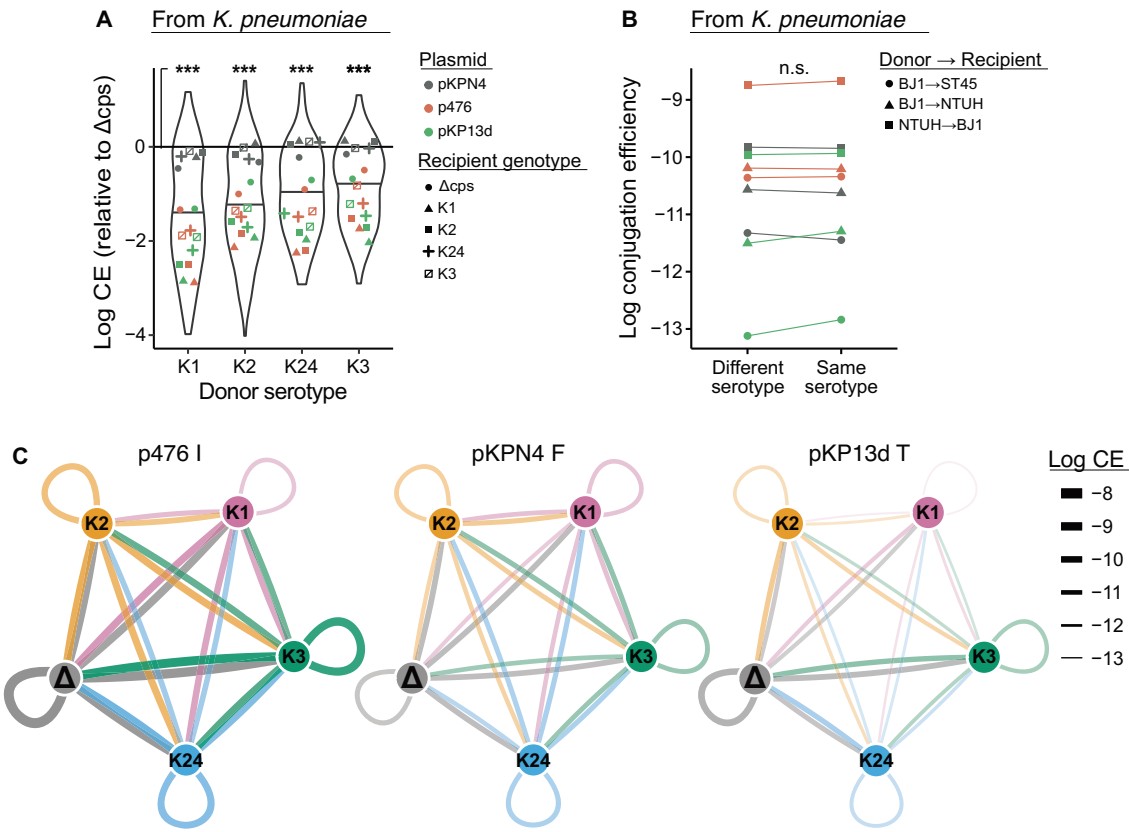

**Fig. 3 | Donor serotype influences conjugation efficiency. A** Log$_{10}$-transformed conjugation efficiency (CE) relative to the associated Δcps mutant (by subtraction of the log$_{10}$-transformed value) by capsule serotype of the donor. Points represent the mean of independent triplicates after transformation and subtraction, with colours and shapes corresponding to strains and plasmids in Fig. 2A. Solid line at $y = 0$ represents the conjugation efficiency of Δcps mutant. All four serotypes are associated with significantly lower conjugation efficiencies than the Δcps mutants (Pairwise Wilcoxon (two-sided), all $p < 0.001$), but not when considering pKPN4 individually (all $p > 0.05$). Individual biological replicates are presented in Supplementary Figs. S4 and S5. **B** Log$_{10}$-transformed conjugation efficiency between pairs of donors and recipients with similar or different serotypes. Points represent mean conjugation efficiencies ($n = 4$ biologically independent experiments), colours indicate different plasmids (as in Figs. 2 and 3) and shapes correspond to distinct pairs. Conjugation is not more efficient between donor and recipient expressing the same serotype, paired Wilcoxon test (two-sided), $p = 0.9$. **C** Networks of plasmid transfer between capsule states. Data drawn from the *K. pneumoniae* to *K. pneumoniae* assays. Nodes represents distinct serotypes, and Δcps mutants. Edges thickness represents the mean conjugation efficiency for all pairs. Edges are coloured according to the donor, indicating the direction of transfer. Source data are provided as a Source Data file 4 (Conjugation from *K. pneumoniae*) and Source Data file 5 (Conjugation within serotype). ***$p < 0.001$; ns $p > 0.05$.

the capsule of the donor (including Δcps). This analysis was only done in the E2 set, since in E1 the donor is an *E. coli* strain lacking a capsule. The conjugation efficiency of Δcps cells was significantly higher than that of any of the four serotypes (Fig. 3A). This difference was significant for plasmids pKP13d$_T$ and p476$_I$, but not for pKPN4$_F$. For the latter, the rates of conjugation seem independent of the presence of a capsule in the donor (just like above they seemed independent of the presence of a capsule in the recipient). In the case of the former (pKP13d$_T$ and p476$_I$), the efficiency of conjugation varied significantly between serotypes (Fig. 3A). The rank of the serotypes in terms of efficiency of conjugation of the donor is similar to that identified above for the recipient: K1 < K2 < K24 < K3. Additionally, we tested if the impact the serotype was symmetrical between donors and recipients, e.g., if BJ1:K1→ NTUH:K2 is equivalent to BJ1:K2 → NTUH:K1. We found that conjugation efficiencies were not significantly different when comparing one direction to the other (Wilcoxon Paired test, $p = 0.8$). Hence, conjugation efficiencies in donor and recipient cells are affected in similar ways by the presence of capsules and by their serotypes.

### Assessing the importance of the different variables on conjugation

The results above suggest that several variables affect the conjugation efficiency. Network analyses of conjugation efficiencies across

serotypes confirm the relevance of the plasmid identity, of the serotype, and the similarity between the effects of donor and recipients (Fig. 3B, C). This results in wide differences in conjugation efficiency, i.e., in fast and slow lanes of horizontal gene transfer. This raises the question of the relative importance of each variable in plasmid transfer and how they jointly contribute to explain differences in conjugation efficiency (see Box 1).

We started by assessing if the MPF type is significantly associated with different conjugation efficiencies when accounting for the effect of the plasmid identity. For this, we used only the data of conjugation from *E. coli* (dataset E1) since in the other dataset (E2) we used three plasmids with a different MPF type each and one cannot distinguish between the two effects (MPF and plasmid identity). First, we found that the interactions between the effects of the MPF type and the recipient genotype are not significant (ANOVA, $p = 0.21$, Statistics 2, Text S2). To test the hypothesis that MPF differ in conjugation efficiencies we used a linear mixed model where the MPF type was the fixed effect, and the plasmid identity was the random effect (Statistics 1, Text S2). This allows to test the effect of the MPF type while conditioning for the effect of the plasmid identity. We found that the effect of MPF is significant (Supplementary Figs. S6A, B, C, S7, F test, $p = 0.023$). The comparisons of all pairs using non-parametric tests and Tukey's HSD analyses show that all pairs of MPF are significantly

different and gives an order of efficiency of conjugation $MPF_F < MPF_T < MPF_I$ in both datasets E1 and E2 (Statistics 2, Text S2, Tukey–Kramer HSD test, $p < 0.001$).

We then tested if the MPF type and serotypes affect the conjugation rates. We made a linear mixed model where we put together as fixed effects the MPF type of the plasmid and the serotypes of the donor and recipient, while conditioning (random effects) for the identity of the donor and recipient chassis strains (Statistics 3, Text S2). This analysis was done with dataset E2 (conjugation to and from *K. pneumoniae*), since in E1 there is no variation in the donor. The results showed that all three effects are significant (F tests, all $p < 0.001$) (Supplementary Fig. S6D, E, F). Interestingly all parameter estimates of the fixed effects were also significant. Hence, differences within each group of variables, notably differences between serotypes both among donors and among recipients, were significant (*t*-tests, all $p < 0.05$). We conclude that all three variables – MPF, donor, recipient serotypes - and that the categories within these variables all contribute significantly to explain the variation in conjugation efficiency.

Given that both donor and recipient serotypes are important for conjugation efficiencies, we tested the hypothesis that combinations of serotypes in donor and recipient cells could improve or decrease conjugation efficiency. For example, strains of the same serotype have been hypothesized to engage more efficiently in mating pair formation than pairs of strains with different serotypes[45]. To address this question, we used standard least squares to model the conjugation efficiency in function of the MPF type and the serotypes of donor and recipients (Statistics 4, Text S2). Here, we used only the data of conjugation between *K. pneumoniae* strains (dataset E2, since for dataset E1 the donor is never capsulated). We also added an interaction term between donor and recipient serotypes. This resulted in a significant linear model ($R^2 = 0.43$, $p < 0.001$, F test), where the tests on the three variables revealed significant effect ($p < 0.001$), but the interaction term was non-significant ($p > 0.9$, same test). Accordingly, none of the comparisons between pairs of same vs. different capsule serotype were significantly different (Fig. 3C). Thus, the capsule serotypes of the donor and recipient cells have a significant and independent impact on conjugation. Finally, one can have a coarse estimate of the relevance of the three significant variables by analysing individual ANOVA where each of the variables is fitted to the conjugation efficiency (Statistics 5, Text S2). While all tests were significant, the effect of the MPF type ($R^2 = 0.32$) was much larger than that of the donor serotype ($R^2 = 0.07$), which exceeded that of the recipient serotype ($R^2 = 0.03$). Hence, the MPF type might have a very important impact on conjugation efficiency.

## Capsule volume correlates negatively with conjugation efficiency

Having established the impact of capsule in conjugation, we set up to uncover a mechanism that could explain our observations: (1) higher conjugation efficiency is associated with the absence of capsule, (2) capsule serotypes affect the rate of conjugation, (3) there is no evidence of specific interactions between capsules of donors and recipients, i.e., conjugation efficiency is similar for pairs with similar or different capsules, (4) MPF pili, which are known to differ widely in length, are important determinants of conjugation efficiency. While (2) and (4) might suggest that some sort of receptor in the recipient cell could explain our results, (1) suggests that capsules are not being used as receptors for the pilus, and (3) suggests that specific interactions between the capsules of the donor and recipient cells are not important. We thus reasoned that the physical barrier represented by the capsule may impede cell-to-cell interactions, leading to inefficient mating-pair formation. To test the hypothesis that capsule volume explains differences in conjugation efficiencies, we measured the effective volume occupied by a cell in a colony (Fig. 4A), estimated via standard-colony dissolution into PBS, as a proxy for the thickness of

the capsule. As a control, we also measured capsule quantity using the traditional glucuronic acid quantification method. Both measures were positively and very significantly correlated (Supplementary Fig. S1, Statistics 6e, Text S2, $R^2 = 0.8$).

The effective volume across all three strains and capsule states spanned a range of 10 to 75 $\mu m^3$/CFU and averaged 32 $\mu m^3$/CFU. This is in line with previous estimates of the packing density of *E. coli* colonies grown in slightly different conditions (33 $\mu m^3$/CFU)[68]. As expected, non-capsulated cells have the smallest volume. This is not caused by a growth defect that could lead to small cells, since we have shown that in rich medium non-capsulated bacteria grow faster and should thus be larger once one excludes the effect of capsules[69]. As expected, all four serotypes were associated with significantly higher effective volumes than Δcps mutants. The K1 and K2 serotypes have the largest volume, which fits published data showing that natural strains with these serotypes tend to have very voluminous capsules[57,70]. Serotypes were associated with different effective volumes in the ranking order: K24/K3 < K2 < K1 (Statistics 6a, Text S2). For example, the effective volume of BJ1::K1 cells is on average *ca.* three times larger than that of BJ1::K3 cells. These results confirm that a large fraction of the volume of the swapped strains can be explained by their different capsular types. It could be argued that cell volume may vary for many other reasons than the capsule, even if we are growing strains in the same conditions and these only differ in the capsule locus. We obtained similar trends when we measured the quantity of capsule with the glucuronic acid quantification method (Supplementary Fig. S1, Statistics 6b, c, d, e, Text S2).

We then tested the hypothesis that capsule volume hinders conjugation by computing the association between the effective volume and conjugation efficiency. To do so, we fitted a linear mixed model with the cell volume as a fixed effect and the chassis strain as a random effect. The results show that the association between the capsule volume and recipient conjugation efficiency is very strong once the identity of the chassis is considered (F test, $p < 0.001$). To note, a similar analysis considering the conjugation efficiency and volume of donors (see Inset text Box 1) provided similar results (Supplementary Fig. S9). To visualize the data in a simpler way, we made three linear regressions between the average effective volume and the average conjugation efficiency of recipients (all conjugation assays) for each chassis strain, which showed a good fit (Fig. 4B). We obtained similar results when fitting the conjugation efficiency to the uronic acid quantity associated with each serotype (Supplementary Fig. S10, Statistics 6b,d). We noticed that the effective volume varies between chassis strains of the same serotype (Supplementary Fig. S11) and leveraged these variations to understand if changes in volume caused a change in conjugation. To test this, we accounted for the serotype by adding it as a random effect in the previous model (Statistics 7, Text S2, Supplementary Fig. S11). This model indicates that capsule volume and recipient conjugation efficiency are negatively correlated within serotypes (F test, $p = 0.004$). These results suggest that the volume of the capsule shapes the differences in conjugation efficiency between strains.

Hence, the effects of the chassis and of the capsule volume explain most of the variation in the conjugation efficiency between serotypes. The analysis of the slopes of these regressions shows that when the average effective cell volume decreases by 20 $\mu m^3$ there is roughly a doubling of the conjugation efficiency. These results are consistent with the hypothesis that capsule volume shapes conjugation efficiency.

## The determinants of conjugation shape the distribution of natural plasmids

Our analysis revealed that conjugation efficiency in the laboratory was strongly impacted by capsule expression and the serotypes of both donor and recipient cells, as well as by the MPF type. Do these results

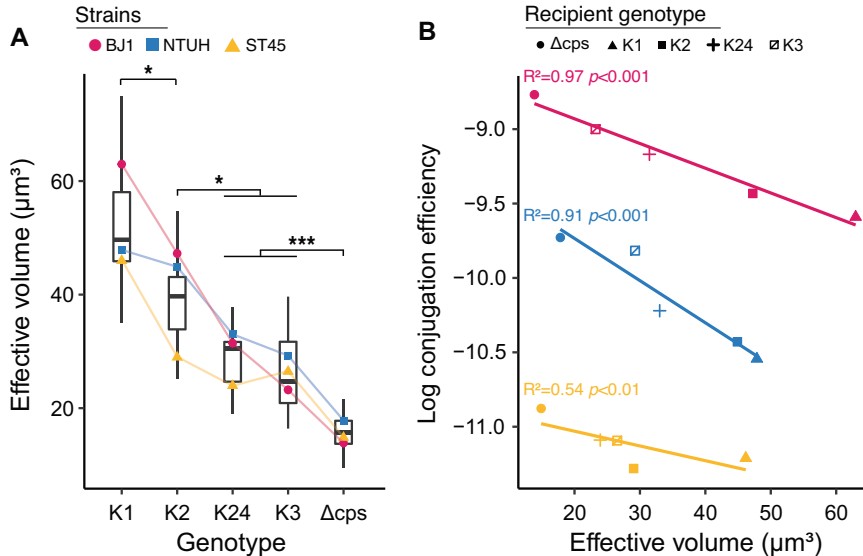

**Fig. 4 | Effective volume within colonies. A** Differences in effective volume (μm³/CFU) between serotype swaps and Δcps mutants. Individual points represent the mean effective volume for each mutant ($n = 4$), while the boxplots are drawn from all the observations. We assessed statistical differences between volumes with an ANOVA followed by a Tukey's honest significant difference (HSD) test. The strain, genotype and their interaction significantly influenced the effective volume. All comparisons between genotypes were significantly different (K3 vs Delta, $p < 0.001$; K2 vs K24, $p = 0.004$; K1 vs K2, $p = 0.004$), except K24 and K3 ($p = 0.85$). Genotypes were ranked according to the median effective volume. Boxes represent the first quartile (Q1), median (line), third quartile (Q3), while whiskers represent the Q1–IQR*1.5 and Q3 + IQR*1.5 (IQR, inter-quartile range). Effective volume positively correlates with amount of capsule quantity (Supplementary Fig. S1A). **B** Mean effective volume vs. mean $\log_{10}$-transformed conjugation efficiencies of recipients (from E1 and E2 combined). Points correspond to distinct capsule states in the recipient strain. Colour corresponds to the strains as in (**A**). Lines represent linear regressions for each chassis strain between the $\log_{10}$-transformed conjugation efficiency and the average effective volume (EV). The linear mixed model of the $\log_{10}$ transformed conjugation efficiency using the effective volume as a fixed effect and the chassis strain identity as a random effect showed a significant effect of the volume ($F$ test, F = 45, $p < 0.001$, Statistics 6, Text S2). Source data are provided as a Source Data file 6 (Volume) and Source Data file 7 (Volume vs. recipient conjugation). *$p < 0.05$; ***$p < 0.001$; ns $p > 0.05$.

contribute to a better understanding of the distribution of plasmids in natural populations? We retrieved the non-redundant dataset of 623 complete genomes of *K. pneumoniae* from RefSeq, which contained 2386 plasmids. We used these genomes to build a pangenome including 29,043 gene families. We extracted the pangenome gene families present in single copy in more than 90% of the strains and used these 3940 gene families to build the species phylogenetic tree and used the genomic data to predict the capsule locus type of each strain (serotype). Finally, each plasmid was characterised in terms of the presence of the conjugation machinery, virulence factors, defence systems, and antibiotic resistance genes. We found that 21% of the plasmids carried all the key genes for conjugation: 23% were of type $MPF_I$, 27% $MPF_T$, and 50% $MPF_F$. Some plasmids of all types carried antibiotic resistance genes (56% of all plasmids) but only Type F plasmids carried virulence factors (8%, half of which carried resistance genes too) (Fig. 5A). Type F plasmids seem to be the least susceptible to the presence of a capsule and they also seem to be the most abundant across the species.

To assess if the rates of plasmid acquisition vary with the serotype, we traced the history of acquisition of each plasmid on the species tree. We found that 68% of the conjugative plasmids were acquired in terminal branches, implying that conjugative plasmids in this dataset were acquired very recently. We then searched to understand if there is an association between the relative frequency of conjugation of each serotype (as measured in the laboratory) and the frequency of acquisition of plasmids by strains in populations of these serotypes. More precisely, we tested if the number of conjugative plasmids acquired on the terminal branches of the tree differed between groups of serotypes. We focused on acquisitions in terminal branches because these are more accurately inferred, and this procedure allows to only count independent events. Among the 77 serotypes present in our genome database, we observed that serotypes differ in the rate of acquisition of conjugative plasmids (Kruskal-Wallis test, $p < 0.001$). We then focused on the serotypes included in this study, namely K1, K2, K3, and K24. The average number of defence systems was not different across serotypes (Statistics 8, $p = 0.22$). Thus, this variable was not considered in further statistical analyses. We pooled the plasmids in two categories because there were only seven K3 genomes: large capsules associated with hyper-virulence (K1, K2) and small capsules (K24, K3). We also put together all plasmids, not just the conjugative ones, because plasmids mobilizable by conjugation will be as affected by the capsule as the conjugative ones. Since we observed a complex relationship between the branch length and the number of plasmid gains (Supplementary Fig. S12), we fitted a generalized additive model with the plasmid gains as response, MPF type and serotype as fixed terms, and the branch length as a smooth term. While there were no significant differences in the types of plasmids acquired between groups, which could be explained by the small size of the dataset ($n = 29$ genomes in the small capsule group), we observed a significantly higher plasmid acquisition rate in genomes of K3 and K24 serotypes than in genomes of K1 and K2 serotypes (Fig. 5B, $p = 0.01$, Statistics 9, Text S2). This is in line with previous genomic analyses showing that the pangenome diversity of hypervirulent strains is lower than that of the multi-resistant ones, mostly due to a lower plasmid diversity[71]. These results suggest that the impact of the capsule serotype on conjugation efficiency plays a role in the differences of rates of plasmid acquisition in natural populations.

## Discussion

Bacteria have multiple mechanisms to restrict infections by MGEs, including intracellular defence systems[72] and surface exclusion systems[73]. They also encode an array of functions with pleiotropic effects on these infections, e.g., involved in DNA repair[74] or present at the cell envelope[75]. Among the latter, the capsule protects from abiotic stresses and the host immune system but is also regarded as a

## BOX 1
# Unravelling the Variables in Bacterial Conjugation

Bacterial conjugation is a multifaceted process, driven by strain-specific factors, plasmid composition, and capsule types. Our multivariate analyses included the following factors:

**Strain attributes:**
- Donor strain: Conjugation efficiency is influenced by the unique traits of the donor. Three specific donors were utilized: *E. coli* DH10B, *K. pneumoniae* BJ1, and NTUH.
- Recipient strain: Similarly, the attributes of the recipient strain play a pivotal role in conjugation efficiency. Three distinct recipients were analysed: *K. pneumoniae* BJ1, ST45, and NTUH.

**Capsule variations:**
- Capsule type: *K. pneumoniae* strains harboured five distinct capsule types: non-capsulated, K1, K2, K24, and K3.

**Plasmid characteristics:**
- Identity: The genetic makeup of a plasmid can determine its impact on conjugation. Ten unique plasmids were studied.
- Mating-Pair Formation (MPF) type: Conjugation system's MPF-type can influence conjugation efficiency. Three different types were studied.

**Analysis framework:**
- Donor's perspective: Evaluates conjugation efficiency by emphasizing the recipient's distinct strain and capsule attributes.
- Recipient perspective: Evaluates efficiency by emphasizing the recipient's distinct strain and capsule attributes.

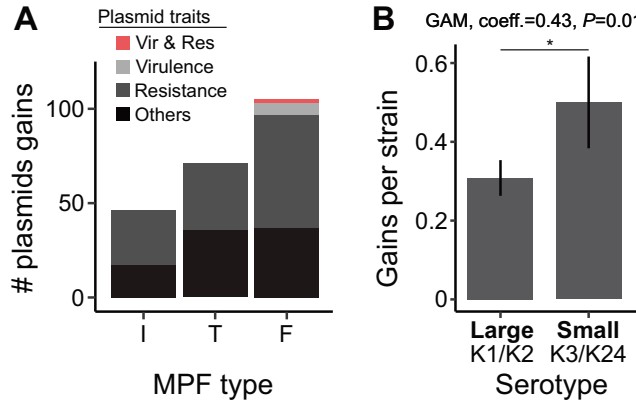

**Fig. 5 | Distribution of recently acquired plasmids.** Analysis of 623 complete K. pneumoniae genomes. **A** Number of conjugative plasmids acquired recently in the genomes of our dataset. The colours correspond to the different categories of plasmids. The data correspond to conjugative plasmids acquired in terminal branches. **B** Number of plasmids (including non-conjugative) recently acquired (terminal branches of the species tree) in the focus serotype. Bars represent the mean of plasmid gains per genome ($n = 73$ for "Large", $n = 29$ for "Small"). Error bars represent the standard deviations. A generalized additive model was fit with the gains as the response variable, $\log_{10}$ branch lengths as a smooth function term (thin-plate regression spline), and serotype (grouped between large and small capsules) and MPF type (I, T, F, Others) as fixed terms (Statistics 9, Text S2). Source data are provided as a Source Data file 8 (Plasmids annotations) and Source Data file 9 (Plasmid acquisitions).

gatekeeper for phages and plasmids. Here we precisely quantified these latter effects in *K. pneumoniae* using isogenic serotype-swap mutants in three different chassis strains. Remarkably, the acquisition of a capsule locus of *ca.* 25 kb in place of the original and its expression did not require additional mutations. This suggests that capsules loci might be transferred easily within the *K. pneumoniae* population where they are ready-to-express, as previously observed in bacteria with high transformation efficiencies such as *Streptococcus pneumoniae*[76]. Such mutants will be useful in the future to probe interactions between capsules, MGEs and other cellular components, e.g., to understand how genetic backgrounds may shape the serotype swaps that hamper capsule-based vaccines long-term efficacy[13,77].

Phage infection requires the availability of a specific receptor at the cell surface. This affects bacterial population dynamics and our ability to leverage phages for antimicrobial therapy. Capsules can mask receptors present on the cell surface, and thus act as barriers to phage infection[1,2,78]. They can also be necessary for phage infection, especially in nearly ubiquitously capsulated species like *Serratia*[79], *Acinetobacter*[21] or *Klebsiella*[24]. The host range of capsule-targeting phages is generally restricted to one or a few serotypes[17,24] because they rely on the tail spike proteins harbouring depolymerase enzymes[19,28]. The latter act as specific receptor-binding proteins tunnelling through capsules[80]. Here, we found that sensitive strains to capsule-dependent phages all became resistant upon capsule inactivation or serotype swap. We also showed that the acquisition of a new serotype resulted in the inversion of phage sensitivity, as swapped strains became sensitive to phages that could infect the original donor of the capsule. Hence, these results

show that independently of the genetic background and differences in the expression of the capsule, the loss or swap of capsules is often necessary and sufficient to change the sensitivity of *K. pneumoniae* to phages.

We examined the influence of the capsule (and its serotype) on conjugation, building on previous findings that showed that the capsule can decrease the rate of acquisition of one specific conjugative plasmid[12]. Our results on a diverse panel of plasmids, donor, and recipient strains show that the presence of capsules in both the donor and recipient cells reduces the efficiency of conjugation (without abolishing it completely). Of note, a recent study on the conjugation of pOXA48 suggests that the effect of capsules may even be stronger in other species, since those of *Klebsiella* were the ones less hindering transfer[81]. Importantly, the number of recent plasmid acquisitions in natural populations matches the estimated rates of conjugation in vitro, i.e., serotypes more permissive to conjugation in the lab correspond to those having acquired more plasmids recently in natural populations. Furthermore, this effect depends on the serotype. Capsules of the K1 and K2 serotypes, which are the most frequent among hypervirulent isolates resistant to phagocyte-mediated clearance[5,82,83], are particularly effective at lowering the rates of conjugation. The capsules of these hypervirulent isolates are often very thick which may facilitate escape to the immune system[57], but also hamper conjugation. Nevertheless, we observed some heterogeneity in the susceptibility to the capsule among the plasmids. For example, pKPN4$_F$ seems to transfer independently of the capsule. This may be due to the presence of the TraNβ allele in the MPF$_F$ apparatus, which was recently shown to stabilize mating-pair formation with *K. pneumoniae* recipients[5,65,82,84] and is present in the MPF system of pKNP4$_F$. Understanding how different factors like TraNβ and the capsule affect conjugation may represent a fruitful venue for future research. Since MPF$_F$ plasmids spread more efficiently than other MPF types in the heavily capsulated cells of hypervirulent strains, this would explain why we found virulence factors exclusively on this type of conjugative plasmids. If so, our results suggest the potential for a trade-off between selection to evade the immune system (with potential implication in virulence) and to

acquire novel traits (with implications in the acquisition of antibiotic resistance) within *Klebsiella* clinical strains. Such a trade-off would be partly alleviated in MPF$_F$ plasmids, which may have contributed to their large success in this species and their unique carriage of both virulence factors and antibiotic resistance genes. The union of the two traits in such conjugative plasmids has led to the recent emergence of hypervirulent and resistant clones[85]. We propose that the ability to conjugate independently of the capsule may be a hallmark of conjugative plasmids conferring hypervirulence and multi-drug resistance.

The effect of capsules on conjugation is intriguing and could arise in multiple ways. We show that it does not depend on interactions between serotypes, suggesting that the effect of capsules is not specifically linked with their composition. Instead, the volume taken by the capsule, and its quantity, seems to explain most of the variation in conjugation rates between otherwise isogenic strains. The capsulated cells occupy more volume than non-capsulated cells in bacterial colonies, and capsule serotypes are associated with different cell effective volumes. For example, K1-expressing cells occupy up to three times more space than non-capsulated cells and conjugate at much lower rates. Moreover, capsule-mediated increase in effective volume is strongly associated with decrease in conjugation efficiency, i.e., the more space a cell takes up, the less efficient it is at conjugation. This suggests that the capsule may affect conjugation by physically distancing the bacteria from each other, making it harder for plasmids to be transferred between them by conjugation. This model can also explain the different interactions between capsules and MPF types, since Type T and I are typically short, non-retractile, and are very much affected by the distance imposed by the capsule. In contrast, the F-pili are longer and can retract, which may allow them to bypass spatial hindrance and could explain why they seem much less affected by the presence of capsules.

Our model describes how phage infection depends on capsule presence and composition, while conjugation efficiency depends on capsule-associated volume for MPF types T and I. Consequently, factors altering the capsule's composition and expression will impact HGT. Although traditionally capsule composition was attributed solely to the genes within the capsule locus, this view has evolved with the discovery of mucoid regulators like *rmpA*[86], and capsule modifiers like prophage-encoded *wzy* able to alter the branching of capsule monomers[87]. In contrast, the expression level of the capsule is known to be influenced by environmental factors such as iron availability or temperature, which are integrated by general regulators to modulate capsule production. Highlighting the connection between capsule regulation and HGT, the Rcs phosphorelay has been observed to co-regulate both the capsule and CRISPR-Cas systems in *Serratia*[88]. Hence, improving our understanding of the regulatory mechanisms of capsule dynamics may provide insights into when and where HGT preferentially occurs.

Transduction, lysogeny and conjugation mediate HGT in most bacterial species. Consequently, factors that influence the rates of infection by phages and conjugative elements shape gene flow[33,62]. In this study, we examined the capsule's dual role in determining phage susceptibility and modulating conjugation efficiency. Our results demonstrate that cell envelope structures play a crucial role in bacterial species evolution. Since capsules are frequently exchanged in natural populations, clones can undergo drastic changes in their ability to access new gene pools following serotype swaps. Changing the capsule type can close or open new routes for phage-mediated HGT and slow down or accelerate plasmid transfer rates. These observations are consistent with previous findings that the transfer of prophages, but not that of conjugation systems, is biased toward same-serotype exchanges[12].

Our results on the role of capsules on conjugation and phage infection can be of relevance to understand other mechanisms affecting capsulated cells and shaping their ecological interactions[89], notably those concerning mechanisms that deliver effectors into other cells. For example, T4SS are part of the MPF leading to plasmid conjugation, but are also used by pathogens to inject proteins and toxins in eukaryotic[35,89,90] and bacterial cells[91]. We suspect that capsules may hinder the attackers and protect the victims from T4SS. Capsules may also offer protection from other syringe-like devices like the type III[92] and type VI[93,94] secretion systems. Accordingly, capsules protect enterobacterial cells, including *K. pneumoniae*, from T6SS killing[95,96]. Extracellular contractile injection systems (eCIS) are toxin-delivery particles that evolved from phages[97]. Capsules might be a protective barrier from eCIS. When the latter specifically target the capsule, like many phages do, then serotype variation may allow bacteria to escape[98]. Capsules may thus impact virulence, population dynamics, bacterial competition, and horizontal gene transfer in multiple ways.

## Methods

### Strains and plasmids
The strains used in this study, as well as their genomic annotations and accession numbers, are described in Supplementary Data S1. The conjugative plasmids used in this study are described in Supplementary Data S2.

### Scarless serotype swaps
The detailed protocol for scarless serotype swaps is available as Supplementary Text 1−Scarless Serotype Swap protocol. Briefly, we first constructed complete capsule loci deletion (Δcps mutants) in our target strains via a Lambda Red knockout with a modified KmFRT cassette (Supplementary Data S3) including an I-SceI cut site outside the Flp recognition target (FRT) sites, and 500 bp of homology upstream *galF* and downstream *ugd*. The KanMX marker was then excised via expression of the FLP recombinase, leaving an 80 bp scar containing the I-SceI cut site between *galF* and *ugd*, which were left intact (Supplementary Table S1, *primers*).

We cloned the whole capsule loci (~25 kb) from strains with distinct capsule serotypes via a Lambda Red gap-repair cloning approach. We cloned capsular loci from the strains involved in the swaps for K2 (BJ1) and K24 (ST45). Capsule locus K1 was cloned from *K. pneumoniae* SA12 (ERZ3205754) which harbour a nearly identical locus as NTUH-K2044 (>99% identity, >99% coverage). Capsule locus K3 was cloned from the reference strain *K. pneumoniae* ATCC 13883.

We built a cloning cassette encoding a KanMX resistance marker, an I-SceI cut site, and low copy pSC101 origin of replication (Supplementary Data S4). This cassette is designed to circularize around the capsule locus, including *galF* promoter and *ugd* stop codon, via recombination to capture the whole locus onto a vector, hereafter referred to as pKapture (Supplementary Table S1, *primers*).

We electroporated pKapture vectors in the Δcps mutants, effectively complementing capsule expression in trans with their own, or other, capsule loci. To force integration of the cloned capsule into its native site, the Δcps strains transformed with pKapture were electroporated with the pTKRED plasmid[99], carrying an inducible I-SceI restriction enzyme, inducible Lambda Red system, and a functional copy of *recA*. Briefly, we induced the I-SceI enzyme overnight with selection to maintain pTKRED. The I-SceI enzyme linearizes the pKapture plasmid, providing recombination proficient linear ends, and introduce a chromosomal double-strand break within the capsule deletion, which is lethal if unrepaired, resulting in the insertion of the capsule locus. When the repair occurs with the linearized plasmid, the I-SceI cut site is removed since the capsule locus recombines outside of the deletion. We identified capsulated colonies on LB plates without selection and sequenced the mutants to validate the proper scarless replacement of the capsule locus. We performed whole-genome sequencing of the swaps by Illumina and used Breseq v0.36[100] to verify

off-target mutations (see *Mutants and transconjugants validation*). All the 12 strains we sequenced carried the expected capsule swap and only one strain carried one missense mutation in the gene *pheP* (NTUH::K24), located outside the capsule locus.

## Uronic acid quantification

The bacterial capsule was extracted as described before[101] and quantified using the uronic acid method[102]. Briefly, OD600nm of overnight cultures in LB medium was measured. Then, 500 µL were transferred to an Eppendorf tube with Zwittergent and were centrifuged. The supernatant was discarded as to specifically measure cell-bound surface polysaccharides, and washed with ethanol, and dissolved in double-distilled water. The uronic acid concentration of each sample was determined from a standard curve of glucuronic acid. Finally, to normalize for number of cells, the concentration was then divided by the OD600nm of the overnight culture.

## Phage assays

**(i) Phage details.** Phage lysates of phK1, phK2 and phK24 were obtained from various laboratories and streaked for single plaques on lawns of the wildtype strains NTUH (phK1), BJ1 (phK2), and ST45 (phK24). Phage phK1 was described in ref. 103. Phages phK2 and phK24 were isolated from sewage from wastewater treatment plants in Valencia (Spain). The former was isolated using *K. pneumoniae* B 5055 capsular type KL2 as host, and phK24 was using *K. pneumoniae* 1680/49 capsular type KL24. Both strains were purchased from the Statens Serum Institute (Copenhagen). For phage isolation, sewage samples were filtered and tested on soft agar semi-solidified media containing a lawn of the *K. pneumoniae* strains.

To isolate each phage, a triple plaque-to-plaque transfer was carried out. Subsequently, each isolated plaque was used to infect log-phase *K. pneumoniae* cultures, and the supernatants were titrated by the standard plaque assay.

**(ii) Phage production.** After overnight incubation, three independent plaques were picked for each phage with a sterile tip and co-inoculated with a single colony of the host strain in 5 mL of fresh LB at 37 °C. The co-culture was spun down the next day and the supernatant was filtered through 0.22 um. 10 uL of each filtrate was introduced in exponentially growing cultures (OD = 0.4) of its corresponding host. Complete lysis was evident for all three phages after 3 h. To recover phage particles, the cultures were centrifuged at 3220 g. Supernatants were mixed with chilled PEG-NaCl 5X (PEG 8000 20% and 2.5 M of NaCl) through inversion. Phages were allowed to precipitate for 15 min and pelleted by centrifugation 10 min at 3220 g at 4 °C. The pellets were dissolved in TBS (Tris Buffer Saline, 50 mM Tris-HCl, pH 7.5, 150 mM NaCl).

**(iii) Phage infections.** To test the susceptibility of wildtype, Δcps mutants, and capsule swapped strains to phages, overnight cultures in LB of strains were diluted 1:100 and allowed to grow until OD = 0.8. 250 µL of bacterial cultures were mixed with 3 mL of top agar (0.7% agar) and poured intro prewarmed LB plates to generate the bacteria overlay. Plates were allowed to dry before spotting serial dilutions of PEG-precipitated phages. Plates were incubated at 37 °C for 4 h and plaques were counted. The lysate titre is defined as the concentration obtained by estimating the plaque forming unit per ml on the lawn of the strain used to prepare the phage lysate.

**(iv) Phage adsorption.** To test the ability of a phage to adsorb, we mixed 10uL of phage stock solution with 500 uL of overnight bacterial culture for 5 min 37 °C. We then placed the tubes in ice, and transferred them in a pre-chilled centrifuge at 4 °C. After centrifugation for 5 min 3220 g, we spotted serial dilution of the supernatant on bacterial lawns of the phage host for PFU counting.

## Selection and characterization of conjugative plasmids

**(i) Plasmid identification.** We screened the genomes of our laboratory collection and the isolates of the National Reference Centre (Centre National de Reference) for Carbapenemase-producing Enterobacteriaceae at the Bicêtre Hospital (Paris) to identify contigs resembling conjugative plasmids. We used Plasmidfinder[104] to retrieve plasmid contigs, MacSyFinder with TXSScan models[64,105] to identify conjugation operons and annotate their mating-pair formation type, and ResFinder to annotate antibiotics resistance genes[106]. We gathered a list of contigs containing the following features: a plasmid replicase identified and typed by PlasmidFinder, a complete conjugation system, and at least one selectable antibiotic resistance (carbapenem or kanamycin resistance, absent in our swapped strains). Additionally, we included two extensively studied conjugative plasmids, pOXA48-K8 (MPF$_I$) and the de-repressed version of the R1 plasmid, R1 *dr19* (MPF$_F$)[107]. We purified our plasmids of interest by conjugation into *E. coli* DH10B, which does not encode any prophage, plasmid, restriction-modification system, and is resistant to streptomycin and auxotroph for leucine. We picked one transconjugant per plasmid and sequenced it to validate that they only contained a single conjugative system (see *Mutants and transconjugants validation*). Those plasmids are described in Supplementary Data S2, and their annotated sequence are available as Supplementary Data S5.

**(ii) Plasmid annotation.** To detect conjugative systems and infer their MPF types, we used TXSScan with default options[35,108]. To detect and annotate virulence factors, we used Kleborate v2.2[109] with default options. To detect and annotate antibiotics resistance genes, we used ResFinder v4.0[106]. We classed plasmids into distinct categories, "resistance plasmids" containing at least one ARG, "virulence plasmids" containing at least one virulence gene, "virulence and resistance plasmids" containing at least one virulence and one resistance gene, and the rest of the plasmids as "others".

**(iii) TraN and TraN receptors typing.** We gathered the sequences of TraN proteins described in ref. 65 (Supplementary Data S6) and searched for homologs among the MPF-F plasmids proteins with Blastp v2.10.0 + [110]. The best hits based on the bitscore of the alignment are displayed in Supplementary Table S2. OmpK36 proteins were identified as the best hit (bitscore) of a Blastp search (Supplementary Table S2) with WP_004180702 (NCBI accession) and aligned with Clustal Omega (Supplementary Fig. S8). The L3 and L4 regions are highlighted according to[65]. OmpA homologs were found through a Blastp search (Supplementary Table S2) with NP_415477 (NCBI accession), and the best hit (bitscore). All three OmpA are 100% identical.

## Mutants and transconjugants validation

We performed DNA extraction with the guanidium thiocyanate method[111] with modifications. DNA was extracted from pelleted cells grown overnight in LB supplemented with 0.7 mM EDTA and appropriate antibiotics for plasmid maintenance. Additionally, RNAse A treatment (37 °C, 30 min) was performed before DNA precipitation. Each clone was sequenced by Illumina with 150pb paired-end reads from NextSeq 550, yielding approximately 1 GB of data per clone. The raw data were deposited in the BioProject PRJNA952961.

All mutants generated in this study were verified by whole-genome sequencing and comparison to the reference wildtype genome with Breseq v0.37.0 with default options[100] (Supplementary Data S7). We also assembled the genomes de novo with Spades v3.15.5[112] (option -*isolate*) and ran Kaptive[48] to annotate and extract the inserted capsule locus. Annotations corresponded with the expected insertion, and alignment of the extracted capsule locus to the source strain with EMBOSS Needle[113] revealed no mutation.

All DH10B transconjugants used as donors were also verified by whole-genome sequencing (Illumina paired-end 300 bp reads), to

verify that only the target plasmid and no other MGEs were transferred. To do so, we assembled the genomes with Spades v3.15.5[112] (option *-isolate*) and analysed the assembly to find plasmid contig(s) as described above in *Selection and characterization of conjugative plasmids*. We found only one plasmid contig per assembly. We used Blastn v2.10.0 +[110] to check that the plasmid contig in DH10B matched one contig from the original donor strain, which was the case. Finally, we extracted and circularized the plasmid contig in DH10B with SnapGene (Supplementary Data S5), according to (i) the paired-end read mapped with bwa-mem v0.7.17[114] and visualized in IGV[115] (ii) the assembly of the plasmid contig in the original donor strain and (iii) long-reads obtained via low-coverage (10x) Pacbio sequencing of the original donor strain that were mapped onto the assembly with bwa-mem v0.7.17 (BioProject: PRJNA952961).

## Conjugation assays

**(i) Experimental setup.** *E. coli* donors were cultured overnight from a single colony with the appropriate antibiotic in 3 mL LB at 37 °C. The next day, cultures were prepared from a 1:50 dilution. *K. pneumoniae* donors and recipients were inoculated over day from single colonies into 3 mL fresh LB with the appropriate antibiotics to avoid the emergence of non-capsulated cells that can appear rapidly under laboratory conditions[69]. We used ertapenem with a final concentration of 0.15 μg/ml for plasmids encoding a carbapenemase and kanamycin with a final concentration of 50 μg/ml for plasmids encoding aminoglycoside resistance genes (Supplementary Data S2).

Cells reached an OD600 ≈ 1 after 4 h of over-day growth, point at which donor cultures were centrifuged and resuspended in 3 mL phosphate-buffered saline (PBS). They were then mixed 1:1 (vol:vol) and a 15 μL drop of the mixture was inoculated on a 24-well microtiter plate containing 1 mL LB agar pads. The droplets were allowed to dry under the hood with laminar flow (5-10 min) and incubated for 1 h in a humidity box at 37 °C. Then, 1 mL PBS was added in each well, and the plates were sealed with a hydrophobic adhesive film and shaken at 120 rpm for 5 min to resuspend the lawn. The contents of each well were then transferred to a 96-well plate, serially diluted and spotted (10 μL) on plates selecting for either donor cells, recipient cells, or transconjugants. The next day, colonies were counted at the appropriate dilution (between 3 and 30 colonies per spot).

**(ii) Selective plating.** We used a selective plating strategy to enumerate exclusively the donors, recipients and transconjugants. To do this, we leveraged the natural markers of our focal strains. This strategy is recapitulated in Supplementary Table S3.

**(iii) Conjugation efficiency estimation.** To measure the transfer of conjugative plasmids, we computed the transfer rate constant[116], or conjugation efficiency[117], with the following method:

$$\text{Conjugation efficiency} = \frac{T}{DR} \cdot \triangle t$$

Where $T$ is the transconjugants concentration (CFU/mL), D the donors' concentration, R the recipients concentration and $\triangle t$ is the time of conjugation. This quantity is expressed in mL.CFU$^{-1}$.hours$^{-1}$ and represents the transfer rate constant[66]. This method performs accurately to estimate the efficiency of conjugation, especially under short conjugation time which minimize the impact of transconjugant conjugation. It was shown to produce consistent estimates of the D:R ratio when compared with far more complex population-based methods[66,67].

We also compared this quantity with the widely used, and simpler formula T/R, and found a Pearson correlation coefficient of 0.97 ($p < 0.001$) between the two quantities after Log$_{10}$-transformation. Hence, these two values are highly correlated and may be interchangeable under our conditions. All analyses were performed with the conjugation efficiency formula above, since it was shown to be less susceptible to experimental parameters[66,67].

## Effective volume measurement

We inoculated a 15 μL drop from an overnight culture for each strain onto LB agar plates and incubated the plates at 37 °C. After 24 h of growth, we resuspended each colony into 1.5 mL Eppendorf tubes containing 50 μL of PBS. Tubes were vortexed for 1 min, then left for 15 min at room temperature, and vortexed again for 1 min. This allowed for complete dissolution of colonies. We then used a 0.5-10 μL micropipette to measure the excess volume. For example, if we measured a volume of 85 μL after colony dissolution, we estimate the colony volume to 85 − 50 = 35 μL. In parallel, we performed serial dilutions followed by plating on LB agar to estimate the total number of CFU in the resuspended colony. The effective volume was then computed as the ratio of CFU by the volume of the colony. This quantity represents the average volume occupied by a cell in a colony. The data are available in Source Data File 6 (Volume). Volume measurements of colonies were performed on the same growth conditions (LB agar at 37 °C) as the conjugation assays.

## Genome data

We retrieved all the *K. pneumoniae* complete genomes available in the NCBI non-redundant RefSeq database, accessed in March 2021, along with their gene annotations. This resulted in a set of 730 genomes containing 2386 associated plasmids. The pairwise genetic distances between all genomes of the species was calculated using MASH[118]. Strains that were too divergent (MASH distance >6%) to the reference strain or too similar (<0.0001) to any other strains were removed from further analysis. A total of 623 genomes were analysed. To identify defence systems, we ran DefenseFinder v1.0.9 with default options[63]. The information on the genomes (including accession numbers) is available in Supplementary Data S8.

## Pan and persistent genomes

The pangenome is the full repertoire of homologous gene families in a species. The pangenome of *K. pneumoniae* was identified using the module pangenome of the software PanACoTa[119] v1.3.1. Briefly, gene families were built with MMseqs2 v13.45111, with an identity and bi-directional coverage threshold of 80%. This analysis resulted in 29,043 gene families among the 623 genomes (Supplementary Data S9). We then computed the persistent genome with a persistence threshold of 90%, meaning that a gene family must be present in single copy in at least 90% of the genomes to be considered persistent, and found 3940 gene families.

## Phylogenetic inference

To compute the species phylogenetic tree, we aligned each of the 3940 protein families of the persistent genome individually with the align module of PanACoTA. These alignments were concatenated to produce a large alignment matrix with 296,147 parsimony-informative sites over a total alignment of 3,740,313 bp. We then use this alignment to make the phylogenetic inference using IQ-TREE (v2.02). We used ModelFinder[120] and calculated 1000 ultra-fast bootstrap[121]. The best-fit model was a general time-reversible model with empirical base frequencies allowing for invariable sites and discrete Gamma model with 4 rate categories (GTR + F + I + G4). We rooted the phylogenetic tree with the *midpoint.root()* function from the Phangorn R package[122]. The tree is available as Supplementary Data S10.

## Capsule locus typing

We used Kaptive v2.0.0[48,123] with default options and the "K locus primary reference" to identify the capsule locus type (CLT) of strains. The predicted CLT is assigned a confidence level, which relies on the overall alignment to the reference CLT, the allelic composition of the locus, and its fragmentation level. We assigned the CLT to "unknown" when the confidence level of Kaptive was indicated as

"none" or "low," as suggested by the authors of the software. The annotation is available in Supplementary Table S8.

Capsule regulators of BJ1, NTUH-K2044 and ST45 were detected according to a previous analysis[124]. Briefly, we compiled a dataset of proteins known to affect capsule production and performed a BLASTP search (Blast+ v2.12)[110] (default parameters) against the full proteome. Proteins that shared more than 80% identity with proteins from the reference dataset were considered as capsule related or belonging to the capsule operon (Supplementary Data S11).

### Reconstruction of the scenario of plasmid gain

We used inference of ancestral states to identify the branches in the tree where the plasmid was acquired. For this, we inferred the ancestral state of each plasmid pangenome family with PastML (v1.9.23)[125] using the MAP algorithm and the F81 model. For each of these gene families, we obtained a list of branches where they were acquired. Genes from the same plasmid can come out of this analysis has having been acquired at different branches if there were changes in the plasmids after acquisition (gene gains or gene losses). To identify the plasmids most likely acquired in the terminal branches of the of the species tree, we counted how many genes of the plasmid were acquired in these branches. We defined that a plasmid was acquired in the terminal branch if at least 75% of its genes were inferred to be absent in the first parental node (i.e., acquired since then). Those are referred as recent gains. The other plasmids were not considered in the analysis presented on Fig. 5. Plasmids annotations are presented in Supplementary Data S12, and plasmid acquisitions are available in Source Data file 9 (plasmid acquisitions).

### Statistics and reproducibility

All the data analyses were performed with R version 4.2 and Rstudio v2022.02.1, except linear mixed models which were computed in JMP v16 (SAS corporation) and generalized additive linear models which were computed with the mgcv R package v1.9 (function gam). For data frame manipulations, we also used *dplyr* v1.0.10 along with the *tidyverse* packages[126] and *dat*a.table v1.12.8. We used the packages *ape* v5.3[127], *phangorn* v2.5.5[122], and *treeio* v1.10[128] for the phylogenetic analyses. We typically show single points when there are less than ten observations, boxplots between 10-100, and violin plots when there are over 100 observations because they might reveal multi-modal underlying distributions. No statistical method was used to predetermine sample size. No data were excluded from the analyses. The investigators were not blinded to allocation during experiments and outcome assessment.

### Reporting summary

Further information on research design is available in the Nature Portfolio Reporting Summary linked to this article.

## Data availability

Source data are provided with this paper. Mutant strains, transconjugants and plasmids sequence generated in this study were deposited in the BioProject PRJNA952961. All data generated in this study have been deposited in the public repository Zenodo https://doi.org/10.5281/zenodo.10635324. Source data are provided with this paper.

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

## Acknowledgements

We thank Olivier Tenaillon and Nienke Buddelmeijer for fruitful discussions. We thank Sylvain Brisse for providing us with the *Klebsiella* strains. We thank Jean-Marc Ghigo and Christophe Beloin for the gift of plasmid R1-*dr19*. We thank Julia Bos for the gift of phage phK1. We are grateful to the members of the Microbial Evolutionary Genomics lab and Maxime Policarpo, Grégoire Haouy and Léone Debarge for helpful discussions. This work used the computational and storage services (TARS & MAESTRO cluster) provided by the IT department at Institut Pasteur, Paris, and the media preparation services provided by the Plateforme Milieux. The sequencing work was performed by the Biomics Platform, C2RT, Institut Pasteur, Paris, France, supported by France Génomique (ANR-10-INBS-09) and IBISA. This work was supported by an ANR JCJC (Agence national de recherche) grant [ANR 18 CE12 0001 01 ENCAPSULATION] awarded to O.R. and by the grant [ANR 16 CE15 0022 03 PREDIRES] awarded to EPCR. The laboratory is funded by a Laboratoire d'Excellence 'Integrative Biology of Emerging Infectious Diseases' grant [ANR-10-LABX-62- IBEID], the INCEPTION programme [PIA/ANR-16- CONV-0005], and the FRM [EQU201903007835]. M.H. has received funding from the FIRE Doctoral School (Centre de Recherche Interdisciplinaire, programme Bettencourt) to attend conferences. P.D.-C. was financially supported by a Ramón y Cajal contract RYC2019-028015-I funded by MCIN/AEI/10.13039/501100011033, ESF Invest in your future. The funders had no role in the study design, data collection and interpretation, or the decision to submit the work for publication.

## Author contributions

M.H., O.R. and E.P.C.R. designed the project. O.R. and E.P.C.R. were involved in the supervision of M.H.; M.H. designed the experimental strategy, built the mutants, and performed the experiments, sequence assembly and comparative genomics. J.L.B. performed the glucuronic acid quantifications. A.N. performed the phage adsorption experiments. R.A.B. collected and provided curated and sequenced bacterial strains. P.D.C. collected and provided curated and sequenced bacteriophages. M.H. and E.P.C.R. performed the statistical analyses. M.H., O.R. and E.P.C.R. wrote the manuscript. All authors provided critical feedback and helped shape the research, analysis and manuscript.

## Competing interests

The authors declare no competing interests.
