## [Peer Review File · Nature Communications]

REVIEWER COMMENTS

Reviewer #1 (Remarks to the Author):

Capsules are a fundamentally important structure, regulating interactions between bacterial cells and the external environment, and limiting access to the cell membrane. However, understanding how diverse capsule structures (serotypes) modulates access of mobile genetic elements (MGEs) such as phages and conjugative plasmids - which are often responsible for transfer of infection relevant traits such as virulence and antimicrobial resistance (AMR) - is difficult due to multiple confounding factors such as the presence of highly diverse defence systems.

Here, this complexity is addressed by constructing a set of *Klebsiella* strains varying capsule serotypes across 3 distinct isogenic backgrounds to create a factorial design of 'swapped' capsules with which to study the interactions between capsules and MGEs. The extensive strain construction work (capsule loci are ~25-30kb) and rigorous statistical analysis enable the authors to disentangle the multi-layered effects which drive variation in plasmid conjugation rates. Whilst plasmid identity effects appear to be driven by mating pair formation (MPF) machinery, overall, there are significant effects of both donor and recipient capsule serotype. Further, significant differences in plasmid acquisition across different serotypes in natural bacterial population (RefSeq database of 623 strains) suggest that a better understanding of the interactions between plasmid MPF types and capsule serotypes will help predict spread of AMR-related traits in natural populations.

This is a very interesting study, thoroughly conducted and well-written, I have only minor comments, mainly asking for clarification.

Minor comments:

L234/Fig2/Fig3. E2 set of conjugation experiments – is there any independent replication within this dataset? Obviously, this is a large-scale experiment therefore there are feasibility issues, and as such am requesting clarification rather than suggesting further experimentation needed.

Difficult to assess variation across replicates/within treatments, error bars would be helpful (e.g., Fig 3B/4B).

L341-369. Is there a significant interaction between MPF and recipient/donor serotypes? i.e., FigS5 suggests some interaction, i.e., I+T MPF types act differently across recipient serotypes, though this may not be significant.

L398. Does effective volume relate to the recipient/donor/interaction? This is unclear to me, and would benefit from clarification.

Fig3B. Does this data include all data from E1 and E2? If so, would benefit from error bars to assess variation and indicate source of data (i.e., E1 vs E2).

L320. This section includes a few different models (e.g., linear mixed effects) - interpretation would be helped by giving supplementary table with model effects (fixed and random), including statistical output (e.g., overall model significance and values for fixed effects and interactions).

L454-9. This suggests that differences in plasmid acquisition rates between serotypes K1/2/3/24 are not significantly different due to low sample sizes (please report stats), but above (L451-2) indicates that there is significance when more serotypes are included in this analysis – these data are not shown?

L757-8 “We then used a 0.5-10ul micropipette to measure the excess volume”. I don’t follow methodology, please expand.

Reviewer #2 (Remarks to the Author):

Haudiquet et al report the importance of the Klebsiella capsule for efficient phage infection and conjugation of plasmids. This was achieved by production of capsule-deletion strains and complementation with different biosynthesis clusters that yield capsules of different presumed composition. The conclusion being that capsule types that produce a higher effective cell volume, from which it is implied a thicker capsule, are less susceptible to phage infection and conjugative events. The experimental data was then validated by sequence analysis of available Klebsiella strains and associated plasmids in the RefSeq genome database.

Overall, the manuscript is well written and concise. The experimental approaches are well designed, and analysis is thorough. However, while the capsule swap section provides interesting observations particularly regarding conjugation efficiency (both for the donor cell and the recipient), the fact that capsule acts as a receptor for bacteriophages has long been known and the data presented is insufficient to justify the mechanistic conclusions. For instance, several of the conclusions depend on correlation data (of serotypes vs plasmid carriage) and the “effective cell volume” approach, which provides a vague estimate at best and does not directly measure the most pertinent factor i.e. the thickness of the capsule.

Specific comments

Abstract

Page 1 Line 23: incur fitness costs, delete “in”

Lines 24-26: “importance [of the capsule] has been difficult to quantify and characterize because of the high diversity of bacterial genomes regarding confounding mechanisms such as antiviral systems”. I do not think this is the best example; anti-viral systems typically work after infection of the cell, i.e., downstream of any capsule effect and thus can be disentangled with more straightforward methods. Something like surface receptor modification would be better here.

Graphical Abstract

This should be completely understandable as a standalone item, however, several elements are not explained e.g. what is setE1 and setE2, what are the K# annotations in the central box, etc..

Introduction

Line 45: Not always the outermost structures, slime layers? Also, other protrusions could be considered more outer e.g. flagella or pili

Line 77: “phages can integrate into the bacterial genome”

Line 101: “varies a lot in terms of volume and chemical composition”, it would be good to briefly summarize this in terms of volume range or typical chemical variants.

Various minor grammar errors, requires careful proofreading

Results

Line 181: “First, the phages could depend on a secondary receptor. We find this hypothesis unlikely given that Δ cps mutants of susceptible hosts were resistant” – I do not think this hypothesis can be ruled out so readily. It is possible that capsule is the primary receptor and is prerequisite for the secondary binding. Further the assertion that the presence of phage defense mechanisms is likely to be the reason for absence of infection is highly speculative, move this to discussion. This section is not actually too important given that 4 out of 6 capsule exchanges were successful.

Line 385: how does the “effective volume” translate to capsule thickness? The hypothesis is that different pili are different lengths and that thicker capsule may prevent function by keeping them out of range, but the volume data do not illustrate that

Line 225: assays not essays

Fig. 4 and 5: Boxplots are insufficiently described e.g. in 5 presumably the grey dots are all individual points overlaid? How many data points are there for each category? It looks like K3 is only ~7 compared to much higher numbers for K1/2. The central line is the median?

Methods:

Not convinced by the accuracy of the “Effective volume measurement”, it is at best a very rough estimate of cell size. Also, I presume the aim was to assess the thickness of the capsule, but total cell volume does not specifically measure that. If the cells are under stress, desiccated, etc then the cell size/morphology could change. Why is no data presented to show cell morphology e.g. microscopy, flow cytometry etc. Could negative staining or an immunological approach have been used to determine the capsule thickness? EM approaches to show the phage or pili kept away from the outer membrane etc.

Emulsification of a colony is a very different condition to the liquid phase phage adsorption assays, the spot pfu assays on soft agar overlays and the LB lawn conjugation assays – is the capsule likely to be the same under all handling conditions?

Unless I missed something, the Methods are incomplete: Glucuronic acid quantification test protocol? Isolation of conjugative plasmids? The “novel scarless method (See Construction of mutants)” is mis-cited and is actually “Supplementary Text 1 - Scarless Serotype Swap protocol”.

Reviewer #3 (Remarks to the Author):

This manuscript reports an extensive study examining the connections between capsule type and different kinds of horizontal gene transfer in *Klebsiella pneumoniae*. This is an extremely important research question as clinically problematic strains of this species possess horizontally-acquired virulence or AMR determinants; understanding their transmission has both fundamental and applied value. The question of how capsules influence HGT (both by transduction and conjugation) has already been addressed in the literature. This study is a substantial advance on previous work because of the scale of the analysis, and the use of isogenic capsule swap mutants allowing effect of capsule type to be queried independently of other factors. The manuscript is well-written and all necessary data are provided. My only major concern is that the effect of capsule amount has not been adequately investigated or considered (expanded on below), however if this issue can be resolved I believe this paper will be suitable for Nature Communications.

The following suggestions need to be addressed before publication:

1. The authors have not measured capsule amount with enough precision, and have not considered the impact of capsule regulation on HGT (either in the experiments reported or in the discussion). This is an important oversight as capsule expression is not expected to be constant in different conditions, different genetic backgrounds or even within a population. My suggestions to address the following are:
 - a. Supplement the colony volume measurements with a more established method to measure capsule amount, such as microscopic measurements or uronic acid quantification, and perform this quantification for each of the capsule-swapped strains. I am not convinced of the reliability of the colony volume measurements as the vortexing during processing may cause capsule shearing, and I am not aware of any independent validation of this method. Each capsule quantification method has limitations so I do not mean for the authors to necessarily repeat the analysis in Fig 4A using a different measure, but at the least the volume measurements need to be validated. As an additional check it could also be useful to compare expression of native and swapped capsule loci in each strain by qRT-PCR.
 - b. Independently test the effect capsule amount on conjugation, for example by using mutants in capsule upregulators such as *rmpA*, or by repeating a subset of the experiments following growth in conditions that produce less capsule (eg. growth at 25 degrees instead of 37 degrees).
 - c. Include information on predicted capsule regulation in each strain, ie. presence/absence of known horizontally-acquired capsule regulators.
 - d. Consider capsule expression in the discussion – HGT in “the wild” is likely to be very significantly influenced by the amount of capsule produced in different conditions and this effect should be considered.
2. The graphical abstract summarises the types of experiments done, but not the findings. I suggest changing to one that outlines the conclusions of the study rather than the methods.

And the following changes would improve the manuscript:

1. In Figure 5, can the authors add a panel that splits the plasmid acquisition events by both MPF type and serotype? This is relevant to the finding that F-type plasmids are less sensitive to capsule volume than others.

2. A very interesting extension experiment would be to test the effect of the TraN-beta allele on conjugation in the context of differently-capsulated recipients and/or recipients with different OmpK36 types, since at the moment it is not clear if the capsule-independence of pKPN4F is due to its length or this specific interaction. Maybe as a simple way to address this the authors could try conjugation with a couple of strains with different OmpK36 sequences?

3. In Figure 1A there is a typo in "mannose"

4. The colour scale for Figure 1B is unclear

5. I could not find the method for uronic acid quantification referenced in Supp Fig 1 legend.

Reviewer #4 (Remarks to the Author):

The manuscript by Haudiquet et al is a detailed investigation of the effect of *K. pneumoniae* capsules on the rates of mobile element transmission. They have engineered novel mutants with altered capsules to test how these structures affect the movement of phage and plasmids. The topic is interesting and the experimental work is very extensive and well-designed.

This work builds on the previous Haudiquet et al manuscript (<https://doi.org/10.1371/journal.pbio.3001276>), which found that:

- "capsules drive phage-mediated gene flow between closely related serotypes"
- "conjugative plasmids are acquired at higher rates in natural isolates lacking a functional capsular locus"

The work also relates to that of Low et al (<https://doi.org/10.1038/s41564-022-01146-4>) on plasmid transfer between Enterobacteriaceae, which concluded that:

- "TraN-OM [an aspect of MPF type] receptor pairings have real-world implications as they reflect the distribution of resistance plasmids within clinical Enterobacteriaceae isolates, demonstrating the importance of mating pair stabilization in mediating conjugation species specificity."

The novelty of this submission is reflected by two statements in the Abstract, which is what I will focus my comments on:

- "Capsule types also affect conjugation efficiency in both donor and recipient cells depending on the serotype, a mechanism shaped by the capsule volume and depending on the structure of the conjugative pilus."
- "Comparative genomics confirmed that more permissive serotypes in the lab correspond to the strains acquiring more conjugative plasmids in nature."

Firstly, regarding the genomic analysis:

1. To analyze the rate of plasmid acquisition in natural populations, the authors infer the number of plasmids acquired on the terminal branches of a species-wide tree. There is no statistical test of whether the trend across the four experimentally-studied serotypes is reflected in natural populations, due to the

small number of KL3 datapoints. However, there is no correction for the length of the terminal branches - these vary extensively across the tree (provided as Dataset S5). It would be expected that more plasmids would be acquired on longer branches - many of the instances of zero acquisitions likely represent very short terminal branches, due to the uneven sampling of the set. There is also no correction for the type of plasmid being acquired, as the experiments suggest that the F MPF type should not vary between capsule types, but the others should (Figure S5). The authors should look at the rate of acquisitions relative to branch length across MPF types for each serotype, perhaps using a linear mixed model, as for the experimental data. This model could then be compared to simpler versions leaving out individual terms to understand what affects the distribution of plasmids in this species.

2. The authors suggest "the existence of a trade-off between virulence...and evolvability". However, *K. pneumoniae* is widespread in the environment, so "virulence" only applies to a limited proportion of the population. Similarly, the KL3 capsule seems to have the highest evolvability, yet is rare in the population. As noted by the authors, "each strain encoded four to eight known defence systems", which would indicate selection against acquisition of plasmids (or evolvability). Outside of clinical isolates, are most plasmids beneficial to *K. pneumoniae*? Could the authors explain how evolvability is compatible with the distribution of defense systems?

Secondly, regarding the experimental studies:

3. The fundamental question I found difficult to answer is how important capsule type is relative to MPF type? Figure 5 suggests MPF is important in natural populations, and a linear mixed model (see above) would help address that point. Similarly, the modeling of the experimental data found MPF had a bigger effect than donor or recipient capsule. It would be helpful if the authors could provide a simple quantification of the relative importance of the two effects. If plasmids are able to survive in the *K. pneumoniae* population with substantial differences in transmissibility, then how much of a barrier are the different capsule types?

4. The authors use a linear mixed model to analyze the data, which seems like a very appropriate approach. However, no formal structure of the model is defined, and there is no comparison of observed and model predicted values, to understand how well the model fits the data.

5. There is no formal test for overfitting of the model. The model should be compared to nested, simpler models (e.g. dropping the terms for donor and recipient capsule type) and comparing these to the evaluated model using a suitable statistic, such as BIC. This would test whether these terms significantly improve the model fit.

6. The effects of different variables are shown in Figure S4. However, there is no representation of the split of data by capsule type, which should be displayed, as this is the main result on which the authors focus.

7. The parameter estimates of the model are not shown. They should be tabulated or plotted, to allow readers to understand the impact of different factors.

8. The strains are included as random effects, which is intuitive, but with so few datapoints, it would make sense to use fixed effects instead (<https://pubmed.ncbi.nlm.nih.gov/35116198/>)

9. Figure 4 shows that the K2 capsule in ST45 has a similar thickness to the K3 capsule in NTUH and the K24 capsule in BJ1. This provides an opportunity to understand whether capsule structure or thickness is important. Does a thinner layer of K2 capsule have less of an effect on conjugation than a thicker layer? How much does the thickness vary for a given capsule type, and how important is that in interpreting the data on the effect of capsules on plasmid acquisition in natural isolates? How big are these differences in capsule thickness relative to the variation in *K. pneumoniae* capsule thickness with regulation (e.g. <https://doi.org/10.1128/mbio.01863-18>)?

My final major comments relate to the presentation of the data, which I found challenging to interpret:

10. Datapoints typically represent “the average of independent triplicates”. Please specify what type of average, and add supplementary plots showing all raw data (i.e. all replicates) to help the reader evaluate the level of variation between assays in Figure 2. Same applies to panel 3A and Figure 4.

11. Figure 2 is difficult to understand. Panel A does not show capsule types. Panel B separates data by recipient strain and serotype. Panel C separates data by donor serotype and recipient serotype. It would be helpful to present these data consistently, using additional panels if appropriate.

12. Figure 4 is very confusing. The panels are not labelled. Why are boxplots used rather than the type of plots shown in Figures 1 and 2? What is the difference between the colored points and the outlier points in panel A?

Minor comments:

- "cps" only italicized sometimes
- "Dataset S8" in the legend of Figure 5 should be S10 I think
- "reunion of the two traits" - were they together previously? Maybe "union"?
- Ref. 23 - misformatted

Matthieu HAUDIQUET
Microbial Evolutionary Genomics Unit
28 rue du Dr Roux, 75015 PARIS
matthieu.haudiquet@gmail.com

REVIEWERS COMMENTS

Note: Reviewer's comments are in blue and our answers in black.

Reviewer #1

Capsules are a fundamentally important structure, regulating interactions between bacterial cells and the external environment, and limiting access to the cell membrane. However, understanding how diverse capsule structures (serotypes) modulates access of mobile genetic elements (MGEs) such as phages and conjugative plasmids - which are often responsible for transfer of infection relevant traits such as virulence and antimicrobial resistance (AMR) - is difficult due to multiple confounding factors such as the presence of highly diverse defence systems.

Here, this complexity is addressed by constructing a set of *Klebsiella* strains varying capsule serotypes across 3 distinct isogenic backgrounds to create a factorial design of 'swapped' capsules with which to study the interactions between capsules and MGEs. The extensive strain construction work (capsule loci are ~25-30kb) and rigorous statistical analysis enable the authors to disentangle the multi-layered effects which drive variation in plasmid conjugation rates. Whilst plasmid identity effects appear to be driven by mating pair formation (MPF) machinery, overall, there are significant effects of both donor and recipient capsule serotype. Further, significant differences in plasmid acquisition across different serotypes in natural bacterial population (RefSeq database of 623 strains) suggest that a better understanding of the interactions between plasmid MPF types and capsule serotypes will help predict spread of AMR-related traits in natural populations.

This is a very interesting study, thoroughly conducted and well-written, I have only minor comments, mainly asking for clarification.

We appreciate your thoughtful review and valuable feedback on our manuscript.

Minor comments:

1.1. L234/Fig2/Fig3. E2 set of conjugation experiments – is there any independent replication within this dataset? Obviously, this is a large-scale experiment therefore there are feasibility issues, and as such am requesting clarification rather than suggesting further experimentation needed.

Difficult to assess variation across replicates/within treatments, error bars would be helpful (e.g., Fig 3B/4B).

Each conjugation assay (defined by Donor + Donor Genotype + Recipient + Recipient Genotype + Plasmid groups) was independently performed three times. We decided not to include error bars in the main figures for visibility and clarity purposes. Given this request (and

those of other reviewers), we have now included panels with individual points in the new supplementary figures S4 and S5. We mention this in the Figure legends.

1.2.L341-369. Is there a significant interaction between MPF and recipient/donor serotypes? i.e., FigS5 suggests some interaction, i.e., I+T MPF types act differently across recipient serotypes, though this may not be significant.

This is a good point, and we agree that Figure S5 suggests there is an interaction between MPF and recipient serotype. Indeed, in the following sections, our hypothesis is that pKPN4 behaves differently than the other plasmids because it is an MPF-F type plasmid.

In dataset E1 (conjugation experiments from *E. coli* to *K. pneumoniae*), we can assess the interaction with an ANOVA since there are several independent plasmids with different MPF. However, the interaction term between MPF and recipient genotype is non-significant. This is now briefly mentioned in the results. (L486-487).

	Df	Sum Sq	Mean Sq	F-value	p-value	
Recipient	2	117.16	58.58	111.832	<2,00E-16	***
Recipient Genotype	3	12.30	4.10	7.828	4.62e-05	***
MPF	2	121.27	60.63	115.755	<2,00E-16	***
Recipient Genotype:MPF	6	4.34	0.72	1.381	0.221	
Residuals	330	172.85	0.52			

1.3.L398. Does effective volume relate to the recipient/donor/interaction? This is unclear to me, and would benefit from clarification.

Sorry for the confusion, we have clarified this in the manuscript. The effective volume was measured separately from conjugation assays, and the data presented in the main figure 4A referred to that of the recipient. Considering this comment, we also performed the same analyses from the perspective of the donor cell, and we observe similar results, namely that donor conjugation efficiency negatively correlates with cell size. This is now included in the Supplementary data as figure S9 and mentioned in the text.

1.4.Fig 3B. Does this data include all data from E1 and E2? If so, would benefit from error bars to assess variation and indicate source of data (i.e., E1 vs E2).

These data come from E2 (*K. pneumoniae* to *K. pneumoniae*), since it's the only set where donors have a K-type. We have added a supplementary figure to show the variability, figure S4 and S5. See also response to comment #1.1.

1.5 L320. This section includes a few different models (e.g., linear mixed effects) - interpretation would be helped by giving supplementary table with model effects (fixed and

random), including statistical output (e.g., overall model significance and values for fixed effects and interactions).

Given this request (and those of other reviewers), we have now included a dedicated section in the Supplementary materials regarding the statistical analysis (Supplementary Text S2). We present the model's significance, and values for fixed/interactions effects. s

1.6 L454-9. This suggests that differences in plasmid acquisition rates between serotypes K1/2/3/24 are not significantly different due to low sample sizes (please report stats), but above (L451-2) indicates that there is significance when more serotypes are included in this analysis – these data are not shown?

We apologize for the confusion. We performed a Kruskal-Wallis on the variable “Serotype” (which includes 75 distinct serotypes) and “Number of plasmids acquired in terminal branches” which is significant and reported in the main text. When we make a subset of our genomic dataset to restrict the analysis to the serotypes K1, K2, K3, and K24, the test is at the edge of significance ($p=0.06$). This may be a consequence of the small number of cases for some serotypes.

We have now clarified this in the main text and added the test value for the subset.

1.7 L757-8 “We then used a 0.5-10ul micropipette to measure the excess volume”. I don't follow methodology, please expand.

We have clarified the method section.

Reviewer #2

Haudiquet et al report the importance of the Klebsiella capsule for efficient phage infection and conjugation of plasmids. This was achieved by production of capsule-deletion strains and complementation with different biosynthesis clusters that yield capsules of different presumed composition. The conclusion being that capsule types that produce a higher effective cell volume, from which it is implied a thicker capsule, are less susceptible to phage infection and conjugative events. The experimental data was then validated by sequence analysis of available Klebsiella strains and associated plasmids in the RefSeq genome database.

Overall, the manuscript is well written and concise. The experimental approaches are well designed, and analysis is thorough.

Dear Reviewer,

Thank you for your thoughtful review and valuable feedback. We are glad you found the manuscript well-written and the experimental approaches well-designed.

2.1 However, while the capsule swap section provides interesting observations particularly regarding conjugation efficiency (both for the donor cell and the recipient), the fact that capsule acts as a receptor for bacteriophages has long been known and the data presented is insufficient to justify the mechanistic conclusions. For instance, several of the conclusions depend on correlation data (of serotypes vs plasmid carriage) and the “effective cell volume” approach, which provides a vague estimate at best and does not directly measure the most pertinent factor i.e. the thickness of the capsule.

We agree that the fact that the capsule can act as a receptor for bacteriophages is known. We now added a reference to this in the second paragraph of the introduction. Here, we provide experimental evidence that the serotype itself (not only the presence of a capsule) determines the host specificity. More importantly, here we show that acquisition of a new capsule serotype not only results in resistance to phages to which the strain was previously sensitive, but also in novel sensitivity to phages to which it was previously resistant. The latter is a key new finding provided by this manuscript. Finally, the experimental demonstration that the capsule serotype is both a mechanism of resistance and the main receptor was not directly shown.

We also acknowledge the concern about the measurement of effective cell volume. We agree and now provide another more classical way of quantifying the capsule (uronic acid quantification). This is now presented in the new Figure S1. Results are very similar, and conclusions remain unchanged.

Specific comments

Abstract

2.2 Page 1 Line 23: incur fitness costs, delete “in”

We have corrected this.

2.3 Lines 24-26: “importance [of the capsule] has been difficult to quantify and characterize because of the high diversity of bacterial genomes regarding confounding mechanisms such as antiviral systems”. I do not think this is the best example; anti-viral systems typically work after infection of the cell, i.e., downstream of any capsule effect and thus can be disentangled with more straightforward methods. Something like surface receptor modification would be better here.

We agree and have changed this as suggested.

Graphical Abstract

2.4 This should be completely understandable as a standalone item, however, several elements are not explained e.g. what is setE1 and setE2, what are the K# annotations in the central box, etc..

We have changed the graphical abstract to make it clearer.

Introduction

2.5.Line 45: Not always the outermost structures, slime layers? Also, other protrusions could be considered more outer e.g. flagella or pili

We have corrected this to “one of the outermost cellular structures”.

2.6 Line 77: “phages can integrate into the bacterial genome”

We changed to “phages can integrate their genome within the bacterial one”.

2.7 Line 101: “varies a lot in terms of volume and chemical composition”, it would be good to briefly summarize this in terms of volume range or typical chemical variants.

We changed to “in terms of chemical composition between serotypes” only since volume was not systematically assessed in previous studies. We have added the following: “So far, eight distinct monosaccharide have been identified in the 77 chemically characterized serotypes, as well as many modifications such as acetylation and a myriad of glycosidic bonds.”

2.8 Various minor grammar errors, requires careful proofreading

We have carefully proof-read the text.

Results

2.9 Line 181: “First, the phages could depend on a secondary receptor. We find this hypothesis unlikely given that Δ cps mutants of susceptible hosts were resistant” – I do not think this hypothesis can be ruled out so readily. It is possible that capsule is the primary receptor and is prerequisite for the secondary binding. Further the assertion that the

presence of phage defense mechanisms is likely to be the reason for absence of infection is highly speculative, move this to discussion. This section is not actually too important given that 4 out of 6 capsule exchanges were successful.

We agree with the reviewer and have changed the text accordingly. On receptor, we wrote: "In this case, our results indicate that the capsule is the primary receptor, as phages are unable to adsorb to the Δ cps mutants of susceptible hosts."

We have also shortened the part on defense systems and now simply mention that they could play a role in explaining our results:

"Those systems, the absence of a secondary receptor, or both may explain why phages sometimes adsorb without starting productive infections."

2.9 Line 385: how does the "effective volume" translate to capsule thickness? The hypothesis is that different pili are different lengths, and that thicker capsule may prevent function by keeping them out of range, but the volume data do not illustrate that

Given that we compared isogenic mutants that differ only in their capsule serotype (or absence of capsule for Δ cps), we believe the effective volume is a proxy for capsule thickness that takes into account all the parameters at play in a bacterial colony, as described in detail in a previous publication (PMID: 28749935). Such measurements are now further supported by the quantification of the capsule by the traditional uronic acid method (new Figure S1). This provides further evidence to our conclusion that the capsule amount negatively correlates with conjugation efficiency. See also our answer to comment 2.12.

2.10 Line 225: assays not essays

We have corrected this mistake.

2.11 Fig. 4 and 5: Boxplots are insufficiently described e.g. in 5 presumably the grey dots are all individual points overlaid? How many data points are there for each category? It looks like K3 is only ~7 compared to much higher numbers for K1/2. The central line is the median?

We now provide more details on the boxplots in the figure legends. The central line represents the median. The grey points indicate different genomes. And the reviewer is correct, there were only seven K3 genomes available in the analyzed dataset. See also answer to reviewer #1, 1.6.

Methods:

2.12 Not convinced by the accuracy of the "Effective volume measurement", it is at best a very rough estimate of cell size. Also, I presume the aim was to assess the thickness of the capsule, but total cell volume does not specifically measure that. If the cells are under stress, desiccated, etc then the cell size/morphology could change. Why is no data presented to show cell morphology e.g. microscopy, flow cytometry etc. Could negative staining or an immunological approach have been used to determine the capsule thickness? EM approaches to show the phage or pili kept away from the outer membrane etc.

The “effective volume” is herein defined as the average volume occupied by a cell in a colony. By measuring the volume of a colony, and the number of CFU (a proxy for the number of cells), we are indeed measuring this effective volume. While it incorporates many variables including the cell size, we use this value to compare isogenic mutants growing under similar conditions that differ only in their capsule locus. Assuming that the mutants are under similar stress, the differences we observe are due to capsule volume, and hence presumably to thickness.

We carefully considered the reviewer’s suggestions. We believe that immunological methods (Quellung reaction, immunofluorescence) are not adapted to compare different serotypes. Indeed, antibody binding to the capsule results in capsule swelling (Methods in Microbiology, Volume 47, 2020, Pages 17-39), modifying the apparent thickness under negative staining. This phenomenon is dependent on each antibody as it is probably dependent on their affinity, and capsule-specific antibodies only bind specific serotypes. Hence, using four distinct antibodies would add more variability to our test and biases that we cannot precisely control, precluding comparisons between serotypes. To our knowledge, there is no study evaluating the capsule size with a flow cytometer which does not involve antibodies.

To address the reviewer's concerns:

- We have now made it clear in the results that this is measured via colony dissolution.
- Following this comment and reviewer’s #3 comment 1.a., we have measured the capsule of all 18 strains by the standard method in the field, namely the uronic acid quantification. We show that the amount of capsule produced correlates very strongly with the effective volume we have measured (and hence negatively with the conjugation efficiency). (Statistics 6b, c in Sup Text S2)

2.13 Emulsification of a colony is a very different condition to the liquid phase phage adsorption assays, the spot pfu assays on soft agar overlays and the LB lawn conjugation assays – is the capsule likely to be the same under all handling conditions?

Indeed. The reviewer is correct, and we cannot be certain that capsule expression is the same in the different growth conditions (although they are all performed in the same growth medium, LB).

Our work has shown that phage adsorption is mainly determined by the chemical composition of the capsule and the presence/absence of receptors. Here, we do not suggest any link between capsule amount and phage sensitivity (either in terms of adsorption or on pfu tests). Capsule production is relevant in the conjugation assays. Conjugations are performed on agar plates of LB at 37°C. These are the same growth conditions at which we measure effective volume. This is now commented in the methods.

2.14 Unless I missed something, the Methods are incomplete: Glucuronic acid quantification test protocol? Isolation of conjugative plasmids? The “novel scarless method (See Construction of mutants)” is mis-cited and is actually “Supplementary Text 1 - Scarless Serotype Swap protocol”.

We have completed the methods and corrected the mis-citation. *Isolation of conjugation plasmids* referred to the section *Selection and characterization of conjugative plasmids*. This is now amended.

Reviewer #3

This manuscript reports an extensive study examining the connections between capsule type and different kinds of horizontal gene transfer in *Klebsiella pneumoniae*. This is an extremely important research question as clinically problematic strains of this species possess horizontally-acquired virulence or AMR determinants; understanding their transmission has both fundamental and applied value. The question of how capsules influence HGT (both by transduction and conjugation) has already been addressed in the literature. This study is a substantial advance on previous work because of the scale of the analysis, and the use of isogenic capsule swap mutants allowing effect of capsule type to be queried independently of other factors. The manuscript is well-written and all necessary data are provided. My only major concern is that the effect of capsule amount has not been adequately investigated or considered (expanded on below), however if this issue can be resolved I believe this paper will be suitable for Nature Communications.

We're grateful for your review and your recognition of the novelty and importance of our study on the connections between capsule type and horizontal gene transfer in *Klebsiella pneumoniae*. Your feedback on the scale of the analysis and the use of isogenic capsule swap mutants is encouraging. We now provide new data on capsule production of all strains in this study, and we hope this effectively addresses your concerns.

The following suggestions need to be addressed before publication:

3.1. The authors have not measured capsule amount with enough precision, and have not considered the impact of capsule regulation on HGT (either in the experiments reported or in the discussion). This is an important oversight as capsule expression is not expected to be constant in different conditions, different genetic backgrounds or even within a population. My suggestions to address the following are:

a. Supplement the colony volume measurements with a more established method to measure capsule amount, such as microscopic measurements or uronic acid quantification, and perform this quantification for each of the capsule-swapped strains. I am not convinced of the reliability of the colony volume measurements as the vortexing during processing may cause capsule shearing, and I am not aware of any independent validation of this method. Each capsule quantification method has limitations so I do not mean for the authors to necessarily repeat the analysis in Fig 4A using a different measure, but at the least the volume measurements need to be validated. As an additional check it could also be useful to compare expression of native and swapped capsule loci in each strain by qRT-PCR.

We thank the reviewer for the constructive feedback.

We first considered performing the RT-qPCR validation. However, we now believe this is not the best approach. Capsule loci encode at least three independent promoters, and while some of the genes are homologous between serotypes, their sequence identity level precludes the use of the same primers between different loci. We fear this will produce a bias. Furthermore, this would only indicate the level of expression of the genes, not the actual amount of capsule. We thus followed another of the reviewer's suggestions and measured the capsule amount by the uronic acid quantification method to validate the effective cell size measurements. The two

measurements are very strongly correlated and support the impact of capsule amount in conjugation efficiency. (See also response to 2.13). We have now included these results in the manuscript (Figure S1).

b. Independently test the effect capsule amount on conjugation, for example by using mutants in capsule upregulators such as *rmpA*, or by repeating a subset of the experiments following growth in conditions that produce less capsule (eg. growth at 25 degrees instead of 37 degrees).

We considered this carefully, but, with all due respect, we decided not to make these analyses.

- The actual role of *rmp* operon in shaping capsule amount has been debated for some time. It is clearly established that it regulated hypermucoviscosity but it is less clear if it regulates the amount of capsule. Two new studies show that effect of *rmp* operon in hypermucoviscosity is independent of capsule amount (PMID: 37140436, PMID: 37610214).

- Considering alternative temperatures, given that the serotypes used here are related to human infections, results at 25° would not be very relevant to *in vivo* conditions.

- Finally, deleting other capsule regulators would introduce other uncontrolled variables. For instance, Δ *rscB* mutants are known to produce less capsule, but this is strain-dependent (as we have previously shown PMID: 36912665). Most importantly, most capsule regulators are global regulators which impact other important surface structures, like LPS, which could *in fine* affect conjugation rates too.

In brief, we do intend to study in the future the impact of genetic regulation on capsule production on the rates of transfer, but we think these analyses will not strongly affect our conclusions and will require a complex long study.

c. Include information on predicted capsule regulation in each strain, ie. presence/absence of known horizontally-acquired capsule regulators.

We have now included a table of presence/absence of genes known to regulate capsule production (Table S6). We used the database from Nucci, A., Rocha, E.P.C. & Rendueles, O. Adaptation to novel spatially-structured environments is driven by the capsule and alters virulence-associated traits. *Nat Commun* 13, 4751 (2022). <https://doi.org/10.1038/s41467-022-32504-9e>

This reflects the complexity of the capsule regulatory network in this species and highlights the difficulty of appropriately addressing the reviewer's previous comments.

d. Consider capsule expression in the discussion – HGT in “the wild” is likely to be very significantly influenced by the amount of capsule produced in different conditions and this effect should be considered.

We have now included a section in the discussion regarding capsule expression regulation and its potential impact on HGT.

3.2. The graphical abstract summarises the types of experiments done, but not the findings. I suggest changing to one that outlines the conclusions of the study rather than the methods.

We agree, and we provide a new graphical abstract where we emphasize the main results of this study.

And the following changes would improve the manuscript:

3.3. In Figure 5, can the authors add a panel that splits the plasmid acquisition events by both MPF type and serotype? This is relevant to the finding that F-type plasmids are less sensitive to capsule volume than others.

In the analysis presented in Figure 5B, all plasmids were included, not just the one identified as conjugative. This was not clear in the text and is now mentioned and explained. Even using conjugative and mobilizable plasmids the statistical signal is weak. Hence, we cannot make an analysis by MPF type because there are too few of those (especially for K24 and K3). Hopefully, the growth of the databases will allow to test this in the future.

3.4. A very interesting extension experiment would be to test the effect of the TraN-beta allele on conjugation in the context of differently-capsulated recipients and/or recipients with different OmpK36 types, since at the moment it is not clear if the capsule-independence of pKPN4F is due to its length or this specific interaction. Maybe as a simple way to address this the authors could try conjugation with a couple of strains with different OmpK36 sequences?

We agree with the reviewer that the TraN/capsule interaction could play a role in conjugation efficiency, unfortunately this type of experiment would not take into account differences in genetic background, which can result in very different conjugation efficiency. The TraN/capsule integration may be the topic of further studies. This is briefly mentioned in the discussion.

3.5. In Figure 1A there is a typo in “mannose”

Thank you, we have corrected this mistake.

3.6. The colour scale for Figure 1B is unclear

We have re-shaped the scale to illustrate that at higher values, conjugation is more efficient. Regarding the colours on their own, the viridis scales provide colour maps that are perceptually uniform in both colour and black-and-white. They are also designed to be perceived by viewers with common forms of colour blindness.

3.7. I could not find the method for uronic acid quantification referenced in Supp Fig 1 legend.

Sorry for the oversight. We have added a section in the Methods.

Reviewer #4

The manuscript by Haudiquet et al is a detailed investigation of the effect of *K. pneumoniae* capsules on the rates of mobile element transmission. They have engineered novel mutants with altered capsules to test how these structures affect the movement of phage and plasmids. The topic is interesting and the experimental work is very extensive and well-designed.

This work builds on the previous Haudiquet et al manuscript (<https://doi.org/10.1371/journal.pbio.3001276>), which found that:

- "capsules drive phage-mediated gene flow between closely related serotypes"
- "conjugative plasmids are acquired at higher rates in natural isolates lacking a functional capsular locus"

The work also relates to that of Low et al (<https://doi.org/10.1038/s41564-022-01146-4>) on plasmid transfer between Enterobacteriaceae, which concluded that:

- "TraN-OM [an aspect of MPF type] receptor pairings have real-world implications as they reflect the distribution of resistance plasmids within clinical Enterobacteriaceae isolates, demonstrating the importance of mating pair stabilization in mediating conjugation species specificity."

Thank you for the time you've taken to provide detailed feedback on our manuscript. Your acknowledgment of the interest and significance of our study examining the impact of *K. pneumoniae* capsules on mobile element transmission is encouraging. We are pleased that you recognize the extensive and well-designed experimental work that has gone into this investigation.

The novelty of this submission is reflected by two statements in the Abstract, which is what I will focus my comments on:

- "Capsule types also affect conjugation efficiency in both donor and recipient cells depending on the serotype, a mechanism shaped by the capsule volume and depending on the structure of the conjugative pilus."
- "Comparative genomics confirmed that more permissive serotypes in the lab correspond to the strains acquiring more conjugative plasmids in nature."

Firstly, regarding the genomic analysis:

4.1. To analyze the rate of plasmid acquisition in natural populations, the authors infer the number of plasmids acquired on the terminal branches of a species-wide tree. There is no statistical test of whether the trend across the four experimentally-studied serotypes is reflected in natural populations, due to the small number of KL3 datapoints. However, there is no correction for the length of the terminal branches - these vary extensively across the tree (provided as Dataset S5). It would be expected that more plasmids would be acquired on longer branches - many of the instances of zero acquisitions likely represent very short terminal branches, due to the uneven sampling of the set. There is also no correction for the type of plasmid being acquired, as the experiments suggest that the F MPF type should not vary between capsule types, but the others should (Figure S5). The authors should look at

the rate of acquisitions relative to branch length across MPF types for each serotype, perhaps using a linear mixed model, as for the experimental data. This model could then be compared to simpler versions leaving out individual terms to understand what affects the distribution of plasmids in this species.

Regarding the comment about the statistical tests. We performed a Kruskal-Wallis test on the number of plasmids acquired on the terminal branches, which is significant when accounting for all serotypes ($p < 0.001$) but not for the smaller subset ($p = 0.06$), most likely, as the reviewer pointed out, due to low sampling of KL3. This is now clarified in the manuscript. Of note, the small dataset size precludes the analysis using mixed models (see also comment 3.3).

Regarding the branch length correction. We agree with the reviewer that more plasmids could be acquired on longer branches. We have further analyzed the dataset, which shows that the relationship between branch length and number of plasmids acquired saturates rapidly above branches of 0.001 substitution/site (panel A). Taking this as a threshold, we present here the results of plasmids acquired between the focus serotypes (panel B) and the branch length correction (panel C), which do not change the conclusion of our global analysis. To note, these results are dependent on the accuracy of the tree branch lengths, which are not representative of time when HGT rates are high (because of chromosomal recombination) which is very hard to control for in a species-wide tree. We have added the following figure as Supplementary Figure S11.

Figure S11 – Plasmid gains according to branches length. A. Observed plasmid gains vs. branch length (log₁₀-transformed). The blue line is fitted by GAM (ggplot2 geom_smooth(method = “gam”)) and shows the saturation of plasmid gains above length > 0.001 (two outliers long branches > 0.01 were removed). **B.** Similar analysis as figure 5B, but with branch length < 0.001. Number of plasmids recently acquired (terminal branches of the species tree) in different serotypes. Individual points represent distinct genomes. **C.** Number of plasmids recently acquired divided by the branch length of the strain harbouring them in different serotypes. Only branches < 0.001 are considered. Individual points represent distinct genomes.

4.2. The authors suggest "the existence of a trade-off between virulence...and evolvability". However, *K. pneumoniae* is widespread in the environment, so "virulence" only applies to a limited proportion of the population.

This is correct, but we do not claim that this trade-off affects Kpn in the environment. It only affects it when it comes to the evolution of virulence. We have clarified this in the text.

Similarly, the KL3 capsule seems to have the highest evolvability, yet is rare in the population.

While KL3 is rare in the public databases, we believe that this is due to sampling bias and probably does not reflect its natural distribution, which is unknown to the best of our knowledge.

As noted by the authors, "each strain encoded four to eight known defence systems", which would indicate selection against acquisition of plasmids (or evolvability). Outside of clinical isolates, are most plasmids beneficial to *K. pneumoniae*? Could the authors explain how evolvability is compatible with the distribution of defense systems?

- We do not know if most plasmids are beneficial to Kpn outside clinical isolates. This is unknown for most species and a very controversial topic that we do not wish to engage in. Of note, most strains in the database are from clinical isolates.
- Our manuscript is not about defense systems. There is a huge literature on the topic being published now and we prefer to avoid that topic, for which we present no experimental data and no remarkable novel computational data. We have here used three genetic backgrounds and all possible combination of serotypes exactly to control for the variations in the genetic background provided by traits such as defense systems. The trade-off between evolvability and defense systems is extremely interesting, but we feel this is outside the topic of this paper. For interesting discussions on this aspect see Oliveira, PNAS, 2016 or the recent preprint from the Brouns lab <https://doi.org/10.1101/2022.08.12.503731>.

Secondly, regarding the experimental studies:

4.3. The fundamental question I found difficult to answer is how important capsule type is relative to MPF type? Figure 5 suggests MPF is important in natural populations, and a linear mixed model (see above) would help address that point. Similarly, the modeling of the experimental data found MPF had a bigger effect than donor or recipient capsule. It would be helpful if the authors could provide a simple quantification of the relative importance of the two effects. If plasmids are able to survive in the *K. pneumoniae* population with substantial differences in transmissibility, then how much of a barrier are the different capsule types?

There are several points in this comment. We tackle them separately.

What matters most, serotype or MPF? We agree this is an important point. We provide some data on this. First, Figures 2 BC and Figure 3A shows that serotypes matter. Figure 3B shows that combinations of serotype do not matter (i.e. these combinations have no explanatory value). Second, Figures 2 BC and Figure 3A show that serotypes affect most plasmid types, except MPFF. This makes the analysis proposed by the reviewer difficult to do, because the effect of one type of plasmids is different from the others. One should also

note that the relative importance of the type of plasmid and capsules will also depend on the conditions of conjugation, e.g. in liquid the type of plasmid is fundamental because MPFT do not usually conjugate. Finally, there are over 100 serotypes predicted in *K. pneumoniae* and we only analyze four. This is enough to obtain several key results, but we think that it is too soon to make a precise quantification of the relative importance of the serotypes in the light of the MPF.

How much of a barrier are the different capsule types? It was not our intention to claim that capsules are perfect barriers to conjugation. Our data shows clearly that they aren't. This is the same for defense systems, none of them provides a perfect barrier to transfer. Our precise statement is "Our results on a diverse panel of plasmids, donor, and recipient strains show that the presence of capsules in both the donor and recipient cells usually reduces the efficiency of conjugation." We think this accurately reflects our results. We have modified a few statements in the text that could be misleading and suggest that capsules absolutely block conjugation. For example, at the end of the above sentence we added "(without abolishing it completely)" to make this completely clear.

4.4. The authors use a linear mixed model to analyze the data, which seems like a very appropriate approach. However, no formal structure of the model is defined, and there is no comparison of observed and model predicted values, to understand how well the model fits the data.

We agree there were too little data on the statistical part. We have now detailed the statistical analyses in a separate file. Please refer to Supplementary Text S2.

4.5. There is no formal test for overfitting of the model. The model should be compared to nested, simpler models (e.g. dropping the terms for donor and recipient capsule type) and comparing these to the evaluated model using a suitable statistic, such as BIC. This would test whether these terms significantly improve the model fit.

Please see the novel statistical results file which shows tests on the relevance of the variables.

4.6. The effects of different variables are shown in Figure S4. However, there is no representation of the split of data by capsule type, which should be displayed, as this is the main result on which the authors focus.

We have added a representation of our conjugation assays split by capsule type for each plasmid, donor and recipient used. This is displayed in the novel Supplementary Figures S4 and S5.

4.7. The parameter estimates of the model are not shown. They should be tabulated or plotted, to allow readers to understand the impact of different factors.

This is now included as a table, see for example "Statistics 3" in Sup Text S2

4.8. The strains are included as random effects, which is intuitive, but with so few datapoints, it would make sense to use fixed effects instead (<https://pubmed.ncbi.nlm.nih.gov/35116198/>)

We now include this in the Statistical analyses file, see Statistics 3 and Statistics 6 for example. This analysis confirms our previous conclusions.

4.9. Figure 4 shows that the K2 capsule in ST45 has a similar thickness to the K3 capsule in NTUH and the K24 capsule in BJ1. This provides an opportunity to understand whether capsule structure or thickness is important. Does a thinner layer of K2 capsule have less of an effect on conjugation than a thicker layer? How much does the thickness vary for a given capsule type, and how important is that in interpreting the data on the effect of capsules on plasmid acquisition in natural isolates? How big are these differences in capsule thickness relative to the variation in *K. pneumoniae* capsule thickness with regulation (e.g. <https://doi.org/10.1128/mbio.01863-18>)?

This is a good point, and we agree that this provides an opportunity to study the root cause of change in conjugation efficiency between serotypes. We now provide a supplementary figure (Figure S10) showing the conjugation efficiency vs. effective volume between E1 and E2 faceted for all plasmids and serotypes. In this figure, we have highlighted strains ST45:K2, BJ1:K24 and NTUH:K3 which were pointed out by the reviewer. We have also added regression lines to help assess the direction of the effect. This is also commented in the results section.

The by-plasmid graphs show in more detail the relationship between conjugation efficiency and effective volume between the specific plasmids included, making it evident for example that pKPN4 and R1 don't behave in a similar way as other plasmids.

The by-serotype graphs show in detail this relationship within serotypes. We account for the intrinsic conjugation efficiency of each strain/plasmid by normalizing with the non-capsulated mutants. Most variance should then come from differences in volume, and not composition. In fact, we observe that within the same serotype, increasing volume seems associated with decreased conjugation efficiency. In a model with the **normalized conjugation efficiency** as the response variable, the volume as a fixed effect, and the plasmid and recipient serotype as random effects, the volume is still significantly associated with the conjugation efficiency (novel Statistics 7).

My final major comments relate to the presentation of the data, which I found challenging to interpret:

4.10. Datapoints typically represent "the average of independent triplicates". Please specify what type of average, and add supplementary plots showing all raw data (i.e. all replicates) to help the reader evaluate the level of variation between assays in Figure 2. Same applies to panel 3A and Figure 4.

The "average of independent triplicates" is the mean of three values obtained from three replicate experiments and after \log_{10} transformation (when specified). This is now clarified in the legends. We have replaced "average" by "mean". We have also added all the raw data points in faceted panels in Supplementary Figures S4 and S5.

4.11. Figure 2 is difficult to understand. Panel A does not show capsule types. Panel B separates data by recipient strain and serotype. Panel C separates data by donor serotype

and recipient serotype. It would be helpful to present these data consistently, using additional panels if appropriate.

In Figure 2A, we present the conjugation efficiency in the WT strains. We have added their initial serotype on the figure. 2. Panels 2B and 2C cannot be presented the same way, because of the differences between the two experiments. For example, panel B represents conjugation from *E. coli*. Hence, there is no "donor serotype". To present the full data more consistently, we have added the individual points in the supplementary material up for each plasmid, from each donor to each recipient, and all combinations of serotypes. Please refer to Supplementary Figures S4 and S5.

4.12. Figure 4 is very confusing. The panels are not labelled. Why are boxplots used rather than the type of plots shown in Figures 1 and 2? What is the difference between the colored points and the outlier points in panel A?

We are sorry for the oversight. We have now labeled panel B appropriately. We show single points when there are less than ten observations, boxplots between 10-100, and violin plots when there are over 100 observations because they might reveal multi-modal underlying distributions. This is now mentioned in the methods (Data analysis section)

The black dots are now removed, they corresponded to values above $Q3+1.5*IQR$ (automatically added by ggplot2 `geom_boxplot()` function).

Minor comments:

- "cps" only italicized sometimes

Thank you, we have corrected this inconsistency.

- "Dataset S8" in the legend of Figure 5 should be S10 I think

The reviewer is correct. This has been amended.

- "reunion of the two traits" - were they together previously? Maybe "union"?

We have changed this to "union".

- Ref. 23 - misformatted

Thank you, we have corrected this.

REVIEWER COMMENTS

Reviewer #2 (Remarks to the Author):

The current version of the manuscript is substantially improved and my previous comments have been addressed adequately.

Reviewer #3 (Remarks to the Author):

The authors have addressed my only major concern, and made additional changes which have significantly improved the manuscript. On the revised manuscript I have no major concerns, just the following suggestion, to be included at the discretion of the editor.

I appreciate the inclusion of uronic acid quantification in addition to effective colony volume measurements and the demonstration of a moderate correlation between the two (r -squared = 0.8). The correlation is there but not particularly strong, so in my view the manuscript would be further improved by more integration of the new results into the manuscript. Specifically, a supplementary figure reproducing Fig 4B but with uronic acid amounts rather than colony effective volume would be useful to readers. I believe that the colony effective volume is likely to be a function of both capsule amount and hypermucoidity (in their rebuttal the authors point out that it can be complex to disambiguate the two in *Klebsiella pneumoniae*), so analysing conjugation efficiency with each of these measures will provide some important context and detail.

Congratulations to the authors on an interesting and valuable study.

Reviewer #4 (Remarks to the Author):

I thank the authors for their response to my previous comments. The addition of the extra information on the statistical methods, and the new plots showing the detail of the effect of capsule size on plasmid gain rates, are a valuable part of the revised work. Point 4.4 has been addressed; my only remaining concern about point 4.3 is that the values in Text S2 should only be provided to a justifiable number of significant figures.

I have minor concerns about the response to point 4.2, mainly regarding interpretation. I have substantial remaining concerns about the response to point 4.1, where I think the newly-added analysis is more susceptible to false positives than that presented originally. My specific comments are below:

4.1: Regarding the rate of acquisition of plasmids in natural populations. My original concern was that the trends shown in Fig. 5B were driven primarily by many short length terminal branches that represented a period too brief for many, if any, plasmid acquisitions to occur. This is confirmed by the newly added Figure S11. Such branches are not informative about rates of plasmid gain. The tree shown in Dataset S5 suggests there is variation in the distribution of such short branches between clades. Therefore differences in population structure or sampling may cause the observed differences in plasmid

acquisitions per terminal branch. Simply, a higher density of sampling results in more short terminal branches, which means more branches on which no plasmids were gained, which implies a slower rate of evolution given the current approach – but this is all driven by the uneven, non-systematic sampling of publicly-available genome sequences.

As an illustration of this point, as part of their response to a different comment, the authors state they believe that KL3 is “rare in the public databases...due to sampling bias”. Sparse sampling of a large population would result in long terminal branches, due to the divergence between distantly-related, infrequently-collected isolates. This artifact of sampling would inflate the number of plasmid gains per terminal branch for KL3, using the current per-branch analysis. This would mean sampling bias could account for the high levels of plasmid gain inferred for this type, rather than the thinness of the capsule.

This would confound one of the main conclusions of the study. The authors’ response to my suggestion was to fit a generalized additive model (GAM) to the number of plasmid gains relative to branch length, identify a level at which they think the number of plasmid gains per unit branch length “saturates rapidly”, and only analyze branches shorter than the saturation threshold. This is problematic because it accentuates the very short branches in their analysis. These are the least informative of relevant variation in evolutionary rates, because they will always be associated with ~0 plasmid acquisitions, and therefore differences between types will just represent differences in tree structure, not differences in rates of plasmid gains. As can be seen from a comparison of Fig. 5B and Fig. S11B, the new analysis is more strongly biased towards the short branches in the KL1 and KL2 trees, which makes the originally-identified problem worse.

A plot of the number of plasmid gains by branch length has been included as Fig. S11C. However, this is still limited by the focus on short branches. The modal number of plasmid gains on the K1 and K2 (note that these are labeled "KL1" and "KL2" in this plot) terminal branches is zero in panel B. Therefore the ratio of plasmid gains per unit branch length will remain zero, regardless of branch length, in panel C. The analysis instead needs to estimate a rate of plasmid gain relative to substitutions per site for each type.

The solution is highlighted by the authors’ fitting of the GAM to the data. This enables a continuous analysis of plasmid gain rates relative to branch length, which should be less biased by sampling or tree structure. A GAM is a type of general linear model, which can be modified to incorporate additional factors, like a mixed effects model. The authors can refit the GAM, adding in terms corresponding to the four *Klebsiella* types (as fixed effects), and the plasmid types (or just whether the plasmids encode TrN-beta or not, or if they are MPF F or not). As the dataset size is limited by the number of branches, not the number of plasmid gains, it should not be a problem to divide up the data in these ways. This will show whether K24 and K3 gain all, or just some, types of plasmid faster than K1 and K2. Alternatively, a linear model could be used, as in other sections of the Results.

4.2: I remain unconvinced by the authors’ claim of “the existence of a trade-off between virulence...and evolvability”. Their response consisted of clarifying that there is no evidence that virulence, or the ability to gain plasmids, is under selection in the majority of *K. pneumoniae* that are not associated with humans; that there is no evidence of the most evolvable type (K3) being common in the *K. pneumoniae* population; that they do not know if there really is a difference in evolvability between types, because they have not considered the effect of defence systems.

There is little evidence that *K. pneumoniae* disease increases its transmissibility. It is more plausible that the capsule helps the bacteria evade the immune system, regardless of the ability to cause symptoms. Unless the authors have evidence that links disease, transmission and the capsule, they should be cautious in suggesting these bacteria are under selection to be virulent.

Similarly, the results do not provide evidence that there is selection for evolvability. They have not looked at the distribution of defence systems. It is entirely possible that the *K. pneumoniae* with thinner capsules compensate by having more defence systems. If the authors believe that an analysis of defence systems is outside the scope of the manuscript, then they should remove all claims about selection for evolvability and “fast and slow lanes of infection” (e.g. in the Abstract). The current results suggest the capsules inhibit gain of plasmids to different extents, but they cannot be applied to overall rates of evolution without considering other factors that affect the spread of plasmids.

REVIEWER COMMENTS

Reviewer's comments in blue, our answers in black.

Reviewer #2 (Remarks to the Author):

The current version of the manuscript is substantially improved and my previous comments have been addressed adequately.

We thank the reviewer.

Reviewer #3 (Remarks to the Author):

The authors have addressed my only major concern, and made additional changes which have significantly improved the manuscript. On the revised manuscript I have no major concerns, just the following suggestion, to be included at the discretion of the editor.

We thank the reviewer for his help in improving our manuscript.

I appreciate the inclusion of uronic acid quantification in addition to effective colony volume measurements and the demonstration of a moderate correlation between the two (r -squared = 0.8). The correlation is there but not particularly strong, so in my view the manuscript would be further improved by more integration of the new results into the manuscript. Specifically, a supplementary figure reproducing Fig 4B but with uronic acid amounts rather than colony effective volume would be useful to readers. I believe that the colony effective volume is likely to be a function of both capsule amount and hypermucoidy (in their rebuttal the authors point out that it can be complex to disambiguate the two in *Klebsiella pneumoniae*), so analysing conjugation efficiency with each of these measures will provide some important context and detail.

We have drawn a similar figure as Figure 4B with regressions between uronic acid amount as a visual companion to the Statistics 6b and 6d in the Supplementary Text S2. We have now moved this figure in the Supplementary Figures file. This shows that the results are the same independently of the measure used to test our hypothesis (effective volume or uronic acid).

We have added several sentences in the text to refer to these results:

1435-437: "We obtained similar trends when we measured the quantity of capsule with the glucuronic acid quantification method (Figure S1, Statistics 6b,c,d,e, Text S2)."

L447-448: "We obtained similar results when fitting the conjugation efficiency to the uronic acid quantity associated with each serotype (Statistics 6d)."

L600-602: "Instead, the volume taken by the capsule, and its quantity, seems to explain most of the variation in conjugation rates between otherwise isogenic strains"

Congratulations to the authors on an interesting and valuable study.

We thank the reviewer for a thoughtful review of our study which helped improve our manuscript.

Reviewer #4 (Remarks to the Author):

I thank the authors for their response to my previous comments. The addition of the extra information on the statistical methods, and the new plots showing the detail of the effect of capsule size on plasmid gain rates, are a valuable part of the revised work.

We thank the reviewer for helping us improve our manuscript.

Point 4.4 has been addressed; my only remaining concern about point 4.3 is that the values in Text S2 should only be provided to a justifiable number of significant figures.

The Text S2 was a request of this and almost all other reviewers. Since the other reviewers are satisfied with its inclusion, we have decided to leave it as is. We can remove it or simplify it if the editor and the reviewer find that is necessary.

I have minor concerns about the response to point 4.2, mainly regarding interpretation. I have substantial remaining concerns about the response to point 4.1, where I think the newly-added analysis is more susceptible to false positives than that presented originally. My specific comments are below:

4.1: Regarding the rate of acquisition of plasmids in natural populations. My original concern was that the trends shown in Fig. 5B were driven primarily by many short length terminal branches that represented a period too brief for many, if any, plasmid acquisitions to occur. This is confirmed by the newly added Figure S11. Such branches are not informative about rates of plasmid gain. The tree shown in Dataset S5 suggests there is variation in the distribution of such short branches between clades. Therefore differences in population structure or sampling may cause the observed differences in plasmid acquisitions per terminal branch. Simply, a higher density of sampling results in more short terminal branches, which means more branches on which no plasmids were gained, which implies a slower rate of evolution given the current approach – but this is all driven by the uneven, non-systematic sampling of publicly-available genome sequences.

As an illustration of this point, as part of their response to a different comment, the authors state they believe that KL3 is “rare in the public databases...due to sampling bias”. Sparse sampling of a large population would result in long terminal branches, due to the divergence between distantly-related, infrequently-collected isolates. This artifact of sampling would inflate the number of plasmid gains per terminal branch for KL3, using the current per-branch analysis. This would mean sampling bias could account for the high levels of plasmid gain inferred for this type, rather than the thinness of the capsule.

This would confound one of the main conclusions of the study. The authors’ response to my suggestion was to fit a generalized additive model (GAM) to the number of plasmid gains relative to branch length, identify a level at which they think the number of plasmid gains per unit branch length “saturates rapidly”, and only analyze branches shorter than the saturation threshold. This is problematic because it accentuates the very short branches in their analysis. These are the least informative of relevant variation in evolutionary rates, because they will always be associated with ~0 plasmid acquisitions, and therefore differences between types will just represent differences in tree structure, not differences in rates of plasmid gains. As can be seen from a comparison of Fig. 5B and Fig. S11B, the new analysis is more strongly biased towards the short branches in the KL1 and KL2 trees, which makes the originally-identified problem worse.

A plot of the number of plasmid gains by branch length has been included as Fig. S11C. However, this is still limited by the focus on short branches. The modal number of plasmid gains on the K1 and K2 (note that these are labeled "KL1" and "KL2" in this plot) terminal branches is zero in panel B. Therefore the ratio of plasmid gains per unit branch length will remain zero, regardless of branch length, in panel C. The analysis instead needs to estimate a rate of plasmid gain relative to substitutions per site for each type.

The solution is highlighted by the authors' fitting of the GAM to the data. This enables a continuous analysis of plasmid gain rates relative to branch length, which should be less biased by sampling or tree structure. A GAM is a type of general linear model, which can be modified to incorporate additional factors, like a mixed effects model. The authors can refit the GAM, adding in terms corresponding to the four *Klebsiella* types (as fixed effects), and the plasmid types (or just whether the plasmids encode TraN-beta or not, or if they are MPF F or not). As the dataset size is limited by the number of branches, not the number of plasmid gains, it should not be a problem to divide up the data in these ways. This will show whether K24 and K3 gain all, or just some, types of plasmid faster than K1 and K2. Alternatively, a linear model could be used, as in other sections of the Results.

In the previous round of reviewing, we had made all additional analyses requested to provide more solid evidence. Yet, we understand the concerns of the reviewer regarding the impact of variable branch length on the estimated rate of plasmid acquisition. We present a novel analysis in the main result section (Figure 5B, L509-523) where we used a GAM as suggested by the reviewer. To account for the small size of the dataset (7 genomes of K3), we grouped together K1 with K2, and K3 with K24. This is justified by the similarities of the serotypes both in term of capsule volume and quantity, but also in term of association with hyper-virulence. As suggested, we have also included the type of plasmid in our analysis.

The results further reinforced our conclusions. The following is a resume from what has been included as the new Statistics 8, Figure 5B and changes in the result section:

We put together K1 and K2 as strains with **large** capsules (associated with hyper-virulence), and K24 and K3 with **small** capsules (not associated with hyper-virulence). We fit a Generalized Additive Model with the number of plasmids acquired in terminal branch as response, the type of plasmid (MPF F, T, I or Others) and strain serotype as fixed effect, and the log₁₀-transformed branch length as a smoothed function. The intercepts are Serotype_Large and Type_Others. Another model with interaction between serotype category and type of plasmid gave the same results, without any significant interaction and with higher AIC (534 vs 533).

Family: Poisson
Link function: log

Parametric coefficients:

	Estimate	Std. Error	z-value	Pr(> z)	
(Intercept)	-0.3186	0.1347	-2.365	0.0180	*
Serotype_Small	0.4313	0.1745	2.472	0.0134	*
TypeT	-1.7540	0.2553	-6.871	6.38e-12	***
TypeI	-2.8526	0.4199	-6.794	1.09e-11	***
TypeF	-1.6487	0.2442	-6.752	1.46e-11	***

Approximate significance of smooth terms:

	edf	Ref.df	Chi.sq	p-value	
s(log_branch_length)	3.253	4.109	47.12	<2e-16	***

R-sq.(adj) = 0.354 ; Deviance explained = 39.7%

UBRE = -0.19739 ; Scale est. = 1; n = 408

We focused on the serotypes included in this study, namely K1, K2, K3, and K24 and pooled them in two categories because there were only seven K3 genomes: large capsules associated with hyper-virulence (K1, K2) and small capsules (K24, K3). We fitted a generalized additive model with the plasmid gains as response, MPF type and serotype as fixed terms, and the branch length as a smooth term. While there were no significant differences in the types of plasmids acquired between groups, which could be explained by the small size of the dataset (n=29 genomes in the small capsule group), we observed a significantly higher plasmid acquisition rate in genomes of K3 and K24 serotypes than in genomes of K1 and K2 serotypes (Figure 5B, Statistics 8, P=0.01).

4.2: I remain unconvinced by the authors' claim of "the existence of a trade-off between virulence...and evolvability". Their response consisted of clarifying that there is no evidence that virulence, or the ability to gain plasmids, is under selection in the majority of *K. pneumoniae* that are not associated with humans; that there is no evidence of the most evolvable type (K3) being common in the *K. pneumoniae* population; that they do not know if there really is a difference in evolvability between types, because they have not considered the effect of defence systems.

We think there is a misinterpretation of our statement. We don't claim this is true for all *Klebsiella*, only for those where pathogenicity is an important fitness component. Also, we suggest that such trade-off might exist, we don't state it as a fact. The complete, previous citation from the discussion paragraph was:

"If so, our results suggest the existence of a trade-off between virulence (the ability to evade the immune system) and evolvability (the ability to exchange genetic material with other bacteria) that may affect clinical strains. This trade-off seems partly alleviated in MPFF plasmids, which may have contributed for their large success in this species and to their unique carriage of both virulence factors and recently acquired antibiotic resistance genes."

We have removed the word "evolvability", and further toned down the sentence to:

"If so, our results suggest the potential for a trade-off between selection to evade the immune system (with potential implication in virulence) and to acquire novel traits (with implications in the acquisition of antibiotic resistance) within *Klebsiella* clinical strains). Such a trade-off would be partly alleviated in MPFF plasmids, which may have contributed to their large success in this species and their unique carriage of both virulence factors and antibiotic resistance genes. [...]"

There is little evidence that *K. pneumoniae* disease increases its transmissibility. It is more plausible that the capsule helps the bacteria evade the immune system, regardless of the ability to cause symptoms. Unless the authors have evidence that links disease, transmission

and the capsule, they should be cautious in suggesting these bacteria are under selection to be virulent.

We have re-worded and toned down this section (see citation above). We do not suggest that these bacteria are under selection, only that if they are, there might be a selection trade-off.

Similarly, the results do not provide evidence that there is selection for evolvability. They have not looked at the distribution of defence systems. It is entirely possible that the *K. pneumoniae* with thinner capsules compensate by having more defence systems. If the authors believe that an analysis of defence systems is outside the scope of the manuscript, then they should remove all claims about selection for evolvability and “fast and slow lanes of infection” (e.g. in the Abstract). The current results suggest the capsules inhibit gain of plasmids to different extents, but they cannot be applied to overall rates of evolution without considering other factors that affect the spread of plasmids.

Since the reviewer seems to consider the analysis of defense systems essential, we have added a novel analysis of defense systems, which shows that there are no differences in the abundance of defense systems between K1, K2, K3 and K24 strains (One-way ANOVA, F-value=1.4, p-value=0.22). As a result, there is now a statistical argument not to include defense systems in the model. The results were thus left unchanged, but this relevant novel analysis was mentioned.

Taken together, all our analyses point that capsule serotypes have different impact on the conjugation rates of both donors and recipients.

REVIEWERS' COMMENTS

Reviewer #4 (Remarks to the Author):

I thank the authors for the changes they have made in response to my comments. I have only three final minor comments:

(1) Text S2: I have no problem with the analysis presented in this part of the manuscript, just that the values should be presented to an appropriate level of precision, given it is focussed on statistical analysis.

(2) Results: In the added section on p. 16, "explain" should be "explained".

(3) Discussion: In the added section on p. 18, "within Klebsiella clinical strains)" does not need a closing parenthesis.

REVIEWERS' COMMENTS

Reviewer #4 (Remarks to the Author):

I thank the authors for the changes they have made in response to my comments. I have only three final minor comments:

We thank the reviewer for the help in improving our manuscript.

(1) Text S2: I have no problem with the analysis presented in this part of the manuscript, just that the values should be presented to an appropriate level of precision, given it is focussed on statistical analysis.

Thank you, we have now added table and figure numbers and legends in Text S2.

(2) Results: In the added section on p. 16, "explain" should be "explained".

Thank you, we have changed to "explained"

(3) Discussion: In the added section on p. 18, "within Klebsiella clinical strains)" does not need a closing parenthesis.

We have removed the extra parenthesis.